# Fast Rate Bounds for Multi-Task and Meta-Learning with Different Sample Sizes

**Hossein Zakerinia**
Institute of Science and Technology Austria (ISTA)
`hossein.zakerinia@ista.ac.at`

**Christoph H. Lampert**
Institute of Science and Technology Austria (ISTA)
`chl@ista.ac.at`

## Abstract

We present new fast-rate PAC-Bayesian generalization bounds for multi-task and meta-learning in the unbalanced setting, i.e. when the tasks have training sets of different sizes, as is typically the case in real-world scenarios. Previously, only standard-rate bounds were known for this situation, while fast-rate bounds were limited to the setting where all training sets are of equal size. Our new bounds are numerically computable as well as interpretable, and we demonstrate their flexibility in handling a number of cases where they give stronger guarantees than previous bounds. Besides the bounds themselves, we also make conceptual contributions: we demonstrate that the unbalanced multi-task setting has different statistical properties than the balanced situation, specifically that proofs from the balanced situation do not carry over to the unbalanced setting. Additionally, we shed light on the fact that the unbalanced situation allows two meaningful definitions of multi-task risk, depending on whether all tasks should be considered equally important or if sample-rich tasks should receive more weight than sample-poor ones.

## 1 Introduction

In multi-task learning (MTL) multiple related tasks are learned jointly, with the goal of improving generalization performance compared to learning each task separately. As a result, MTL is particularly promising in settings where tasks are individually data-scarce but collectively rich in shared information, such as personalized Internet services, healthcare applications, or autonomous driving. Due to its fundamental nature, aspects of MTL also influence many other recent developments in the field of machine learning, such as distributed learning (Verbraeken et al., 2020), federated learning (Zhao et al., 2018; Kairouz et al., 2021), multi-agent learning (Gronauer & Diepold, 2022) or meta-learning (Hospedales et al., 2021). Besides numerous algorithmic contributions, there is also a rich collection of results on the theoretical properties of MTL, in particular the improved generalization guarantees it provides compared to single-task learning.

Recent works on the generalization properties of machine learning models are often stated in the framework of PAC-Bayesian generalization bounds (McAllester, 1998). In contrast to classical approaches, such as VC-theory (Vapnik & Chervonenkis, 1971) or Rademacher complexity (Bartlett & Mendelson, 2002), these tend to provide tighter results with the potential to not only provide structural insight but even be numerically informative (i.e. non-vacuous) (Lotfi et al., 2022, 2024).

For MTL, a number of PAC-Bayesian generalization guarantees have been derived, often in combination with related results for *meta-learning*. Initially, these were in the *standard-rate* setting (Pentina & Lampert, 2014, 2015; Amit & Meir, 2018; Rothfuss et al., 2023), with convergence rates at best $O(\sqrt{1/M})$, where $M$ is the total number of training data points.

Quite recently, also some *fast-rate bounds for MTL* were developed with convergence rates up to $O(1/M)$ (Guan & Lu, 2022; Zakerinia et al., 2025). However, there is a fundamental limitation: **existing fast-rate bounds for MTL hold only if all tasks have the same number of samples**. This is an unrealistic restriction for practical settings, such as federated learning or healthcare.

In this work, we close this surprising gap in the literature: we prove fast-rate bounds for multi-task learning, for which tasks can have different numbers of samples (called *unbalanced*), for both of the so-called **kl**-style (Seeger, 2002; Maurer, 2004) as well as "Catoni-style" (Catoni, 2007) bounds. However, our contribution is not only technical, but also conceptual: first, we demonstrate that the unbalanced setting is not simply a minor variant of the balanced one, but has its own statistical properties. Specifically, **the results and proofs of the existing fast rate bound do not carry over to the unbalanced setting**, but **fast-rate bounds for unbalanced multi-task learning are possible**. Consequently, for our results, we establish a new path for proving fast-rate bounds that we expect to be of independent interest also for other situations. Second, we identify the previously overlooked fact that the unbalanced setting allows for not just one but two meaningful definitions for the multi-task risk: *task-centric* or *sample-centric*. The task-centric risk assigns equal importance to each task. This makes sense, for example, if tasks are individual customers, and one wants to give each of them the best possible experience. The sample-centric risk assigns weights to tasks proportionally to how many training data points they have. This makes sense when the core aim is to make as many correct predictions as possible, e.g. when tasks are products and samples are individual sales. Finally, we also extend our results to the meta-learning setting (Schmidhuber, 1987; Baxter, 2000).

In summary, our main contributions are **the first fast-rate generalization bounds for unbalanced multi-task learning, both in the task-centric and the sample-centric setting**. As an additional contribution, we provide new insights into the statistical properties of these learning settings, and a numeric analysis of their potential, in particular in comparison to existing standard-rate bounds.

## 2   Background

**Multi-task and Meta-learning.**    Machine learning tasks can be described as triples, $t = (D, S, \ell)$, where $D$ is a data distribution, $S$ is a dataset of $m$ i.i.d. samples from $D$, and $\ell$ is a loss function in $[0, 1]$. Learning a task means finding a model, $f$, from a family of models, $\mathcal{F}$, that has a small risk (expected loss), $\mathcal{R}(f) = \mathbb{E}_{z \sim D} \ell(f, z)$, with respect to the distribution $D$. In standard (single-task) learning, the learning algorithm can only use the dataset $S$ to determine $f$. Typically it does so by minimizing the training risk $\widehat{\mathcal{R}}(f) = \frac{1}{m} \sum_{i=1}^{m} \ell(f, z_i)$, potentially in combination with some regularization terms.

In *multi-task learning (MTL)* (Caruana, 1997), multiple tasks, $t_1, \ldots, t_n$, are given. The goal is to learn individual models $f_1, \ldots, f_n$ for the given tasks, but the learning algorithm can do so jointly, using the data from all tasks simultaneously. The implicit assumption is that the different tasks are related and that sharing information across them could improve performance. This reflects how humans often learn: by leveraging shared structures and knowledge across tasks to learn more efficiently. In this work, we refer to the number of tasks as $n$ and the number of training samples for task $i$ as $m_i$.

*Meta-learning* (Schmidhuber, 1987), or *learning to learn* (Thrun & Pratt, 1998), extends multi-task learning to the setting where, in addition to the observed tasks, there will also be future tasks that are not observed yet. The goal is to use the observed tasks' data to find a learning algorithm $A$ from a set of possible algorithms $\mathcal{A}$ that will learn good models on future tasks. To theoretically study meta-learning, one formalizes the relationship between tasks by assuming the existence of an environment of tasks $\mathcal{T}$ and an (unknown) environment distribution $\tau$, from which both the observed training tasks and future tasks are i.i.d. samples (Baxter, 2000).

**PAC-Bayesian bounds.**    PAC-Bayesian generalization bounds (McAllester, 1998), i.e. guarantees for the generalization gap of stochastic models, have gained significant interest in recent years,

particularly for their ability to provide non-vacuous generalization bounds for neural networks (Dziugaite & Roy, 2017; Lotfi et al., 2022), unlike traditional approaches such as VC dimension (Vapnik & Chervonenkis, 1971) or Rademacher complexity (Bartlett & Mendelson, 2002).

Stochastic models are parametrized as a distribution (also called *posterior*) over the hypothesis set. For any distribution, $Q \in \mathcal{M}(\mathcal{F})$, its loss is the expected loss of models drawn according to $Q$, i.e. $\ell(Q, z) = \mathbb{E}_{f \sim Q} \ell(f, z)$. PAC-Bayes bounds provide upper-bound for $\mathcal{R}(Q) = \mathbb{E}_{z \sim D} \ell(Q, z)$ using the training risk $\widehat{\mathcal{R}}(Q) = \frac{1}{m} \sum_{i=1}^{m} \ell(Q, z_i)$, and a complexity term which usually is based on the Kullback-Leibler divergence between the posterior and a data-independent *prior*.

Standard PAC-Bayesian bounds provide high-probability guarantees that the true risk does not exceed the training risk by more than a term that decreases with a rate of $O(1/\sqrt{m})$. Here is an example of a PAC-Bayes bound:

**Theorem 2.1** ((McAllester, 1998))**.** *For any fixed $\delta > 0$, and data-independent prior $P$, with probability at least $1 - \delta$ over sampling of a dataset $S$ we have*

$$\mathcal{R}(Q) \leq \widehat{\mathcal{R}}(Q) + \sqrt{\frac{\mathbf{KL}(Q\|P) + \log(\frac{1}{\delta}) + \frac{5}{2}\log(m) + 8}{2m - 1}}. \tag{1}$$

To prove this result, and many other PAC-Bayesian results, one follows a common blueprint: 1) apply a *change of measure inequality* (Seldin et al., 2012) to bound the expectation of a function with respect to the posterior $Q$ by the complexity term $\mathbf{KL}(Q\|P)$, and the expectation of a corresponding *moment generating function (MGF)* with respect to the prior $P$, which is independent of training samples. 2) one shows that the MGF for the independent samples is bounded by a sufficiently small term. Often, this involves upper-bounding it by the expectation of an MGF with respect to i.i.d. Bernoulli random variables, and upper-bounding this upper bound by a more-or-less explicit calculation.

Besides standard (also called *slow-rate*) bounds, such as (1), it is also possible to construct bounds that can give tighter guarantees, even a $O(1/m)$ convergence rate when the training error is small. The following theorem is an example of such a *fast-rate* bound:

**Theorem 2.2** ((Maurer, 2004))**.** *For any fixed $\delta > 0$, and data-independent prior $P$, with probability at least $1 - \delta$ over sampling of a dataset $S$ we have*

$$\mathbf{kl}(\widehat{\mathcal{R}}(Q)|\mathcal{R}(Q)) \leq \frac{\mathbf{KL}(Q\|P) + \log(\frac{2\sqrt{m}}{\delta})}{2m}, \tag{2}$$

*where*

$$\mathbf{kl}(q|p) = q \log \frac{q}{p} + (1 - q) \log \frac{1 - q}{1 - p} \tag{3}$$

In fast-rate bounds, the dependence between training risk and true risk is typically implicit, such as characterized by their Kullback-Leibler divergence in (2). However, for any observed $\widehat{\mathcal{R}}(Q)$, one can derive an explicit upper bound on $\mathcal{R}(Q)$ by numerically inverting the $\mathbf{kl}$ expression with respect to its second argument.

The main advantage of fast-rate bounds is that they provide much tighter guarantees in the regime where the training risk is small, as is commonly the case when working with rich model classes such as neural networks. As a consequence, fast-rate PAC-Bayesian bounds are among the most promising tools for studying the generalization properties of deep neural networks (Dziugaite & Roy, 2017; Pérez-Ortiz et al., 2021; Lotfi et al., 2022), even including large language models (Lotfi et al., 2024).

**PAC-Bayesian multi-task learning and meta-learning.**     Following Pentina & Lampert (2014), PAC-Bayes is an important framework to study multi-task learning and meta-learning, as it naturally formalizes information sharing through the concept of a prior.

In PAC-Bayesian multi-task learning, we wish to learn posterior distributions for the $n$ given tasks jointly. However, instead of data-independent priors, we can learn a (distribution over) data-dependent prior to be shared for all tasks. Formally, we learn a shared hyper-posterior $\mathcal{Q}$ over priors, and different

posteriors, $Q_1, \ldots, Q_n$, for different tasks. We define two distributions over $\mathcal{M}(\mathcal{F}) \times \mathcal{F}^{\otimes n}$, i.e. they assign joint probabilities to tuples, $(P, f_1, ..., f_n)$, which contain a prior over models, and $n$ models. A standard PAC-Bayesian multi-task bound then has form

**Theorem 2.3.** *For any fixed hyper-prior $\mathcal{P}$, any $\delta > 0$, it holds with probability at least $1 - \delta$ over the sampling of the training datasets, $S_1, \ldots, S_n$, that for all hyper-posterior $\mathcal{Q}$ and all posteriors $Q_1, \ldots, Q_n$:*

$$\frac{1}{n} \sum_{i=1}^n \mathcal{R}_i(Q_i) \leq \frac{1}{n} \sum_{i=1}^n \widehat{\mathcal{R}}_i(Q_i) + \sqrt{\frac{\mathbf{KL}(\mathfrak{Q} \| \mathfrak{P}) + \log \frac{4 m_h n}{\delta} + 1}{2 m_h n}}, \tag{4}$$

*where $\mathcal{R}_i$ and $\widehat{\mathcal{R}}_i$ are the expected risk and training risk of task $i$, $m_h = \frac{n}{\sum \frac{1}{m_i}}$ is the harmonic mean of the training set sizes, $m_i = |S_i|$. $\mathfrak{Q}$ is a distribution over $\mathcal{M}(\mathcal{F}) \times \mathcal{F}^{\otimes n}$ given by the product $\mathcal{Q} \times Q_1 \times \cdots \times Q_n$, and $\mathfrak{P}$ is a distribution given by the generating process:* i) *sample a prior $P \sim \mathcal{P}$,* ii) *for each task, $i = 1, \ldots, n$, sample a model $f_i \sim P$.*

From the fact that $\mathbf{KL}(\mathfrak{Q} \| \mathfrak{P}) = \mathbf{KL}(\mathcal{Q} \| \mathcal{P}) + \sum_{i=1}^n \mathbb{E}_{P \sim \mathcal{Q}} \mathbf{KL}(Q_i \| P)$ (Lemma B.4 in the appendix) one can see the importance of the hyper-posterior. Compared to a naive bound with $n$ complexity terms based on the individual posteriors and a fixed prior $P$, in (4) the prior can be *chosen in a data-dependent way* (by means of choosing $\mathcal{Q}$), at the expense of only one additional complexity term $\mathbf{KL}(\mathcal{Q} \| \mathcal{P})$. When the tasks are related in a way that allows for a common prior, the multi-task bound can be much tighter than single-task bounds could. See Appendix A for more discussion.

PAC-Bayes *meta-learning* was first formalized in Pentina & Lampert (2014) as transferring the prior learned using the training tasks to future tasks to be used for minimizing a PAC-Bayes bound. This was followed by a rich line of works (Amit & Meir, 2018; Liu et al., 2021; Guan & Lu, 2022; Riou et al., 2023; Friedman & Meir, 2023; Rezazadeh, 2022; Rothfuss et al., 2023; Ding et al., 2021; Tian & Yu, 2023; Farid & Majumdar, 2021; Scott et al., 2024) in different setups. In this work, we follow the general form introduced in Zakerinia et al. (2024), which formulates meta-learning as learning a general stochastic learning algorithm for future tasks. Formally, the goal is to learn a meta-posterior $\rho$ over a set of algorithms $\mathcal{A}$, generating training posteriors $A(S_1), \ldots, A(S_n)$, and using a multi-task hyper-posterior $\mathcal{Q}(A)$ as described above.

## 3 Unbalanced multi-task learning

In this section, we describe our main technical and conceptual contributions. First, we briefly demonstrate in what sense the existing proofs for fast-rate bounds do not carry over from the balanced to the unbalanced multi-task situation. Then, we discuss the requirements that fast-rate bounds should possess in the unbalanced situation, in particular we draw attention to the fact that one has to make a choice whether to bound the task-centric or the sample-centric risk (which we will introduce there). And finally, for both of these, we prove a series of generalization bounds, first for the multi-task learning and then extending them to the meta-learning setting.

### 3.1 Existing proofs do not carry over to the unbalanced setting

Existing fast-rate multi-task bounds (Guan & Lu, 2022; Zakerinia et al., 2025) are proved following essentially the same steps as their single-task analogues. First, one applies a change of measure inequality. Second, one bounds the empirical multi-task risk $\frac{1}{n} \sum_{i=1}^n \frac{1}{m} \sum_{i=1}^m \ell(f_i, z_j^i)$ by an average $\hat{\mu} = \frac{1}{n} \sum_{i=1}^n \frac{1}{m} \sum_{j=1}^m X_{i,j}$, where the $X_{i,j}$ for $i = 1, \ldots, n$ and $j = 1, \ldots, m$ are Bernoulli random variables with a common mean $\mu$. Third, one uses Theorem (1) from Maurer (2004) to establish a bound on the resulting MGF $M_\mu(\lambda) = \mathbb{E}[e^{n\lambda \mathbf{kl}(\hat{\mu} | \mu)}] \leq 2\sqrt{nm}$ for any $\lambda \leq m$. Finally, one uses Markov's inequality and rearranges terms to obtain the final form of the bound, where the multiplier $\lambda n$ in front of the $\mathbf{kl}$-term in the MGF becomes the denominators of the complexity term, i.e. here the rate of convergence is up to $O(1/nm)$.

In the following Lemma we show that this otherwise ubiquitous proof technique fails in the case of unbalanced MTL, because the corresponding MGF does not have a finite bound for sufficiently large multipliers.

**Lemma 3.1.** *Let $X_{i,j} \overset{i.i.d.}{\sim}$ Bernoulli$(\mu)$ for $i = 1, \ldots, n$ and $j = 1, \ldots, m_i$. Define $\widehat{\mu} = \frac{1}{n} \sum_{i=1}^{n} \frac{1}{m_i} \sum_{j=1}^{m_i} X_{i,j}$, as their average, weighted by inverse dataset sizes. Let $m_{min} = \min_i m_i$, Then, the MGF $M_\mu(\lambda) = \mathbb{E}\left[e^{n\lambda \, \mathbf{kl}(\widehat{\mu}|\mu)}\right]$ has the property that $\sup_{0<\mu<1} M_\mu(\lambda) = +\infty$, whenever $\lambda > m_{min}$. In particular, there is no finite bound on $M_\mu(\lambda)$ that depends only on $n$ and the $m_i$ but not $\mu$.*

**Discussion.** A consequence of Lemma 3.1 is that the best multi-task rate achievable by the proof technique above is $O(1/nm_{\min})$. However, the resulting bound would not improve when the number of samples for some tasks increases while $m_{\min}$ remains the same. This is unintuitive and inconsistent with the standard-rate bound in Theorem 2.3, whose convergence depends on the harmonic average of dataset sizes, $m_{\mathrm{h}}$, rather than their minimum. Also, a known property of implicit bounds in the form of Theorem 2.2 is that one can derive corresponding explicit standard-rate bounds from them by means of Pinsker's inequality, $\mathbf{kl}(q|p) \geq 2(q - p)^2$. Doing so in this situation, however, would result in a standard-rate bound with the suboptimal rate $O(\sqrt{1/nm_{\min}})$ instead of $O(\sqrt{1/nm_{\mathrm{h}}})$. Overall, our analysis suggests that a different approach should be considered to achieve fast-rate bound for the unbalanced multi-task learning, and we do so in the following sections.[1]

Note that in this paper, we focus on the fast-rate bounds for unbalanced multi-task learning in the PAC-Bayesian framework, however, a similar problem also exists in the PAC setting, i.e., the current fast-rate bounds in the PAC literature would also have dependency on the minimum number of samples (Yousefi et al., 2018).

### 3.2 Task-centric vs sample-centric guarantees for unbalanced multi-task learning

Unarguably, the goal of multi-task learning is to learn multiple models that perform well for their respective tasks. As such, one might argue that MTL is actually a *multi-objective* learning problem (Sener & Koltun, 2018). In practice, however, one needs to define a scalar objective to quantify the success of learning in general, and generalization in particular (Hu et al., 2023). Most existing work in multi-task learning use a uniform average across tasks for this purpose, resulting in a multi-task risk, and its empirical counterpart

$$\mathcal{R}^{\mathrm{T}}(Q_1, \ldots, Q_n) = \frac{1}{n} \sum_{i=1}^{n} \mathcal{R}_i(Q_i) = \frac{1}{n} \sum_{i=1}^{n} \mathbb{E}_{z \sim D_i} \ell_i(Q_i, z), \tag{5}$$

$$\widehat{\mathcal{R}}^{\mathrm{T}}(Q_1, \ldots, Q_n) = \frac{1}{n} \sum_{i=1}^{n} \widehat{\mathcal{R}}_i(Q_i) = \frac{1}{n} \sum_{i=1}^{n} \frac{1}{m_i} \sum_{j=1}^{m_i} \ell_i(Q_i, z_{ij}). \tag{6}$$

We call this setting *task-centric MTL*, because each task is treated as equally important, regardless of how much data it contributes to the learning. An alternative scalarization is *sample-centric MTL*, in which tasks get weighted proportionally to their amount of training data. The corresponding risk and empirical risk are

$$\mathcal{R}^{\mathrm{S}}(Q_1, \ldots, Q_n) = \sum_{i=1}^{n} \frac{m_i}{M} \mathcal{R}_i(Q_i) = \sum_{i=1}^{n} \frac{m_i}{M} \mathbb{E}_{z \sim D_i} \ell_i(Q_i, z), \tag{7}$$

$$\widehat{\mathcal{R}}^{\mathrm{S}}(Q_1, \ldots, Q_n) = \sum_{i=1}^{n} \frac{m_i}{M} \widehat{\mathcal{R}}_i(Q_i) = \frac{1}{M} \sum_{i=1}^{n} \sum_{j=1}^{m_i} \ell_i(Q_i, z_{ij}), \tag{8}$$

respectively, where $M = \sum_{i=1}^{n} m_i$ is the total number of data points.

In balanced MTL, both notions coincide, so no decision between them is necessary. In unbalanced MTL, however, they are distinct, and we argue that the choice of preferable objective depends on the problem setting. For example, consider a common scenario, in which a service provider trains client-specific models. If the goal is to maximize client satisfaction, the task-centric setting is appropriate. If, however, the goal is make as few mistakes as possible on future data, data-rich clients should get more attention, because they will likely contribute also a larger fraction of the future data points. Consequently, the sample-centric setting is preferable.

---

[1] Note that none of the discussed problems occur for the balanced setting with $m_1 = \ldots, m_n = m$, because there $m_{\min} = m = m_{\mathrm{h}}$.

Consequently, in this work we provide new results for both settings. All proofs are provided in the supplemental material.

### 3.3 Fast-rate generalization bounds for task-centric multi-task learning

Our main results are the following generalization bounds, which establish the first fast-rate generalization guarantees for unbalanced multi-task learning.

**Theorem 3.2.** *For any fixed hyper-prior $\mathcal{P}$, any $\delta > 0$, it holds with probability at least $1 - \delta$ over the sampling of the training datasets that for all hyper-posterior functions $\mathcal{Q}$ and all posteriors $Q_1, \ldots, Q_n$ :*

$$\sum_{i=1}^{n} m_i \, \mathbf{kl}(\widehat{\mathcal{R}}_i(Q_i)|\mathcal{R}_i(Q_i)) \leq \mathbf{KL}(\mathfrak{Q}\|\mathfrak{P}) + \log\frac{1}{\delta} + \sum_{i=1}^{n} \log(2\sqrt{m_i}) \qquad (9)$$

*For any fixed hyper-prior $\mathcal{P}$, any $\delta > 0$, and any $\lambda_1, \ldots, \lambda_n > 0$, it holds with probability at least $1 - \delta$ over the sampling of the training datasets that for all hyper-posterior functions $\mathcal{Q}$ and all posteriors $Q_1, \ldots, Q_n$:*

$$-\sum_{i=1}^{n} m_i \log(1 - \mathcal{R}_i(Q_i) + \mathcal{R}_i(Q_i)e^{\frac{-\lambda_i}{nm_i}}) \leq \frac{1}{n}\sum_{i=1}^{n} \lambda_i \widehat{\mathcal{R}}_i(Q_i) + \mathbf{KL}(\mathfrak{Q}\|\mathfrak{P}) + \log(\frac{1}{\delta}) \qquad (10)$$

**Discussion.** For proving these results (unlike the balanced setting), we consider independent terms capturing the derivations related to each task (which depends on its own sample size). By jointly bounding the combination of these terms to gain the shared complexity term based on hyper-posterior.

The **kl**-type bound (9) relates the individual per-tasks risk with their empirical estimates. For the balanced setting with $m_1 = \cdots = m_n = m$, it recovers previous fast-rate results (up to log terms), because $nm \, \mathbf{kl}(\widehat{\mathcal{R}}^{\mathrm{T}}|\mathcal{R}^{\mathrm{T}}) \leq \sum_i m \, \mathbf{kl}(\widehat{\mathcal{R}}_i|\mathcal{R}_i)$ due to Jensen's inequality. The form we provide also handles the unbalanced case, for which—in light of our discussion in Section 3.1—we avoid having to bound $\mathbf{kl}(\widehat{\mathcal{R}}^{\mathrm{T}}|\mathcal{R}^{\mathrm{T}})$ directly, instead finding the weighted linear combination of per-task **kl**-terms to be a more suitable target.

To better understand the behavior of (9) in the unbalanced setting, assume a setting in which all tasks have fixed training set sizes, except for one, say task $k$, for which we consider larger and larger training set sizes, i.e. $m_k \to \infty$. Then, for identical $(\mathcal{Q}, Q_1, \ldots, Q_n)$, and therefore constant $\mathbf{KL}(\mathfrak{Q}\|\mathfrak{P})$, the left hand side of the bound grows with $m_k \, \mathbf{kl}(\widehat{\mathcal{R}}_k, \mathcal{R}_k)$, while the right hand side grows as $\log(\sqrt{m_k})$, which implies fast-rate convergence $\mathbf{kl}(\widehat{\mathcal{R}}_k, \mathcal{R}_k) \to 0$. In particular, tasks with few samples do not *slow down* generalization of tasks with many samples.

As a second indication that (9) provides the right characterization, observe that it readily implies an explicit standard-rate bound of the correct rate $O(\sqrt{1/nm_{\mathrm{h}}})$, using the relation

$$2\,n\,m_{\mathrm{h}}\Big(\frac{1}{n}\sum_{i=1}^{n}\big(\mathcal{R}_i - \widehat{\mathcal{R}}_i\big)\Big)^2 \;\leq\; 2\sum_{i=1}^{n} m_i\big(\widehat{\mathcal{R}}_i - \mathcal{R}_i\big)^2 \;\leq\; \sum_{i=1}^{n} m_i \, \mathbf{kl}\big(\widehat{\mathcal{R}}_i \,\|\, \mathcal{R}_i\big), \qquad (11)$$

where the left inequality follows from the Cauchy-Schwarz and the right one from Pinsker's inequality.

The Catoni-type bound (10) offers a more explicit characterization of the generalization behavior, because the empirical risk appears explicitly on the right hand side of the inequality. Like the **kl**-bound, it guarantees fast-rate convergence for any task, even if the training set sizes of all other tasks remain fixed.

A natural question is which of the two bounds yields better guarantees. As it turns out, this depends on the choice of $\lambda_1, \ldots, \lambda_n$. It is known (Germain et al., 2009, Proposition 2.1) that $\sup_{\lambda>0}\big[-\log(1 - p + pe^{-\lambda}) - \lambda q\big] = \mathbf{kl}(q|p)$, i.e., an *optimal* choice of the $\lambda_i$ would recover (9), except without the log terms on the right hand side. Unfortunately, (10) does not hold uniformly with respect to the $\lambda_i$-values, so one cannot simply optimize the expressions to obtain the best values. However, if one has a candidate set of potential values, the bound can be made uniform for this set by a union bound, and select the tightest one. This is also the strategy we follow in our empirical evaluation, see Section 5. Note, however, there is no guarantee that the resulting bound will improve over (9), so a promising strategy is to also include that one into the union bound.

## 3.4 Numerical computation of the bounds

Single-task **kl**-bounds are straightforward to compute numerically: because only a single quantity is unknown (the true risk $\mathcal{R}$), one can make use of the fact that $\mathbf{kl}(q|p)$ is strictly monotonically increasing and therefore invertible in $p \in [q, 1]$ to identify the largest values of $\mathcal{R}$ such that $\mathbf{kl}(\widehat{\mathcal{R}}|\mathcal{R})$ fulfills the bound. Similarly, single-task Catoni-style bounds can readily be used to derive a numeric bound on the risk by observing that $-m \log(1 - \mathcal{R} + \mathcal{R}e^{\frac{-\lambda}{m}})$ is a strictly monotonically increasing function of $\mathcal{R}$.

In the MTL setting, the bounds in Theorem 3.2 have $n$ unknowns, $\mathcal{R}_1, \ldots, \mathcal{R}_n$, but the corresponding bounds provide only a single joint constraint. As such, the set of feasible solutions is much richer than in the single-task setting. In the task-centric setting, we are interested in guarantees on the largest possible value for $\mathcal{R}^{\mathrm{T}} = \frac{1}{n}\sum_i \mathcal{R}_i$, i.e. we have to numerically solve the optimization problem

$$\mathcal{R}^{\mathrm{T}*} \quad \leftarrow \quad \max_{\mathcal{R}_1,\ldots,\mathcal{R}_n} \left[\frac{1}{n}\sum_i \mathcal{R}_i\right] \quad \text{subject to generalization bound constraint.} \tag{12}$$

We illustrate this process in Figure 3. Interestingly, and to our knowledge unique to the MTL setting, the potentially complex geometry of the constraint set allows obtaining tighter guarantees by combining multiple bounds. For example, we can instantiate both (9) and (10) and combine them by a union bound. Optimizing over the resulting constraint set can provide an even better guarantee on $\mathcal{R}^{\mathrm{T}}$ than the minimum of using each bound individually. Figure 3 also illustrates this effect, which is impossible in the single-task setting due to the one-dimensional nature of the problem there.

For our experiments, we use Sequential Least Squares Programming (SLSQP) (Kraft, 1988), a gradient-based optimization algorithm that solves constrained nonlinear problems by iteratively approximating them with quadratic programming subproblems. Note that this procedure is computationally inexpensive compared to model training.

## 3.5 Generalization bounds for sample-centric multi-task learning

As discussed in Section 3.2, sample-centric MTL has a different goal than task-centric MTL: its objectives are weighted by the training set sizes, to reflect that tasks with many samples can also be expected to occur more often in the future. In this section, we provide the fast-rate generalization bounds for the corresponding risk (7).

**Theorem 3.3.** *For any fixed hyper-prior $\mathcal{P}$, any $\delta > 0$, it holds with probability at least $1 - \delta$ over the sampling of the training datasets that for all hyper-posterior functions $\mathcal{Q}$ and all posteriors $Q_1, \ldots, Q_n$:*

$$\mathbf{kl}(\widehat{\mathcal{R}}^S|\mathcal{R}^S) \leq \frac{\mathbf{KL}(\mathfrak{Q}\|\mathfrak{P}) + \log\frac{2\sqrt{M}}{\delta}}{M} \qquad \textit{sample-centric } \mathbf{kl}\textit{-bound} \tag{13}$$

*For any fixed hyper-prior $\mathcal{P}$, any $\delta > 0$, and any $\lambda > 0$, it holds with probability at least $1 - \delta$ over the sampling of the training datasets that for all hyper-posterior functions $\mathcal{Q}$ and all posteriors $Q_1, \ldots, Q_n$*

$$\frac{-M}{\lambda}\log(1 - \mathcal{R}^S + \mathcal{R}^S e^{\frac{-\lambda}{M}}) \leq \widehat{\mathcal{R}}^S + \frac{\mathbf{KL}(\mathfrak{Q}\|\mathfrak{P}) + \log(\frac{1}{\delta})}{\lambda} \quad \textit{sample-centric Catoni-bound} \tag{14}$$

**Discussion.** The provided bounds are more similar to balanced multi-task learning, and have the following properties: 1) The focus is on the sample level, and separating terms based on the tasks is not necessary. 2) All samples contribute the same amount to the training risk, and an issue such as in Lemma 3.1 does not happen. 3) The sample complexity is based on the total number of samples, and the minimum sample size or harmonic mean is not a bottleneck.

**Corollary 3.4.** *Applying Pinsker's inequality to the $\mathbf{kl}$-bound of Theorem 3.3 results in the following explicit standard-rate bound for sample-centric multi-task learning:*

$$\mathcal{R}^S \leq \widehat{\mathcal{R}}^S + \sqrt{\frac{\mathbf{KL}(\mathfrak{Q}\|\mathfrak{P}) + \log\frac{2\sqrt{M}}{\delta}}{2M}} \tag{15}$$

# 4 Meta-learning with unbalanced tasks

In this section, we extend our results to meta-learning. Following the framework of Zakerinia et al. (2024), the goal is to learn a *learning algorithm*, $A$, from a candidate set, $\mathcal{A}$, to be used for future tasks. At meta-training time, a set of training tasks are available that were sampled i.i.d. from a task environment that allows for tasks to have different training set sizes in $[1, m_{\max}]$. As in multi-task learning, previous works studied this problem only in a task-centric view. However, it is also possible and relevant to study sample-centric meta-learning for the setting in which tasks with more training samples should get more weight in the analysis. Consequently, we define two meta-learning risks:

$$\mathcal{R}_M^{\mathrm{T}}(\rho) = \mathop{\mathbb{E}}_{A \sim \rho} \mathop{\mathbb{E}}_{(D,m,S) \sim \tau} \mathop{\mathbb{E}}_{z \sim D} \ell(z, A(S)), \quad \mathcal{R}_M^{\mathrm{S}}(\rho) = \mathop{\mathbb{E}}_{A \sim \rho} \mathop{\mathbb{E}}_{(D,m,S) \sim \tau} \mathop{\mathbb{E}}_{z \sim D} \frac{m}{m_{\max}} \ell(z, A(S)), \quad (16)$$

where $\rho$ is a meta-posterior over the set of algorithms $\mathcal{A}$. Each algorithm in $A \in \mathcal{A}$ is a function that given a dataset would generate a posterior distribution $A(S)$. Additionally, each algorithm can learn a hyper-posterior $\mathcal{Q}(A)$ similar to multi-task learning. Therefore, for each algorithm we would use the two distributions. 1) $\mathfrak{Q}(A)$: a hyper-posterior $\mathcal{Q}(A)$ specific to algorithm $A$ (data-dependent) and task posteriors $A(S_i)$, and 2) $\mathfrak{P}(A)$: a hyper-prior $\mathcal{P}(A)$ specific to algorithm $A$ (data-independent) and priors $P \sim \mathcal{P}(A)$.

In this section, we provide fast-rate bounds for unbalanced meta-learning. Here, we state the kl-style bounds, while the analogous Catoni-style bounds and the proofs for both types are provided in Appendix B.4. To provide our results, we first introduce the following notation, which is the upper-bound given numerical optimization of the kl-bound explained in Section 3.4.

$$\mathbf{kl}_{m_1,\ldots,m_n}^{-1}\left(q_1,\ldots,q_n \Big| b\right) = \sup\left\{\frac{1}{n}\sum_{i=1}^n p_i \Big| \sum_{i=1}^n m_i \, \mathbf{kl}(q_i|p_i) \leq b, p_i \in [0,1]\right\} \quad (17)$$

With this notation we provide our main meta-learning bounds.

**Theorem 4.1.** *For any fixed meta-prior $\pi$, and fixed set of $\mathfrak{P}(A)$, any $\delta > 0$, it holds with probability at least $1 - \delta$ over the sampling of the training datasets that for all meta-posteriors, and hyper-posteriors $\mathcal{Q}(A)$:*

*Task-centric meta-learning: with $c_1 = \sum_{i=1}^n \log(2\sqrt{m_i})$*

$$\mathcal{R}_M^T(\rho) \leq \mathbf{kl}_n^{-1}\left(\mathbf{kl}_{m_1,\ldots,m_n}^{-1}\left(\widehat{\mathcal{R}}_1(\rho),\ldots,\widehat{\mathcal{R}}_n(\rho)\Big|C(\rho)+c_1\right)\Big|\mathbf{KL}(\rho\|\pi)+\log\frac{4\sqrt{n}}{\delta}\right), \quad (18)$$

*Sample-centric meta-learning: with $c_2 = \log(2\sqrt{M})$*

$$\mathcal{R}_M^S(\rho) \leq \mathbf{kl}_n^{-1}\left(\frac{M}{nm_{max}}\mathbf{kl}_M^{-1}\left(\widehat{\mathcal{R}}^S(\rho)\Big|C(\rho)+c_2\right)\Big|\mathbf{KL}(\rho\|\pi)+\log\frac{4\sqrt{n}}{\delta}\right) \quad (19)$$

*where $C(\rho) = \mathbf{KL}(\rho\|\pi) + \mathop{\mathbb{E}}_{A\sim\rho}[\mathbf{KL}(\mathfrak{Q}(A)\|\mathfrak{P}(A))] + \log\frac{2}{\delta}$.*

**Discussion.** The meta-learning generalization bounds are based on two parts: 1) a multi-task bound which upper-bounds the generalization error within the training tasks 2) a generalization bound at the environment level, to upper-bound the expected performance for the environment based on the true risk of the training tasks.

Numerically computing the bounds consists of first numerically computing the upper-bound for the true risk of the training risks, and then numerically computing the final bound. Similar to multi-task learning, the bounds have a better sample complexity over the sample size of training task when the risks are small, and additionally, a better complexity over the number of tasks, for example for task-centric bounds, when the risks are small, the rate of the bounds would be $O(1/m + 1/n)$ instead of $O(\sqrt{1/m} + \sqrt{1/n})$. Additionally, applying Pinsker's inequality to the task-centric bound, would result in the standard-rate meta-learning bounds of Zakerinia et al. (2024). Similarly, by using Pinsker's inequality, we get standard-rate bounds for sample-centric meta-learning.

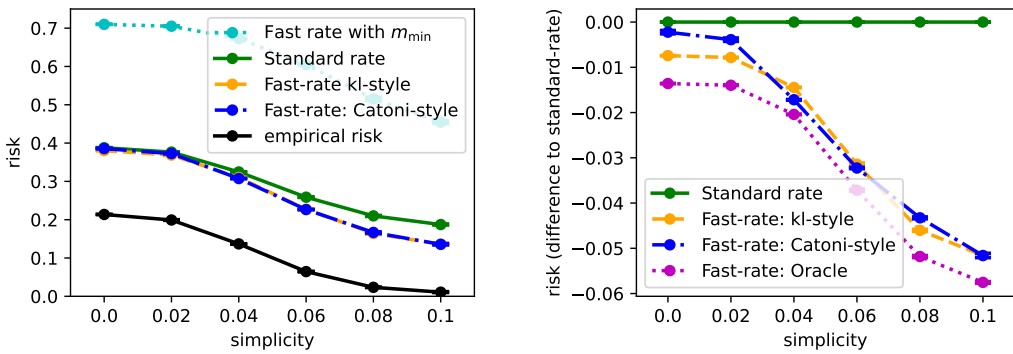

Figure 1: Task-centric MTL: graphical results for linear models on the MDPR dataset

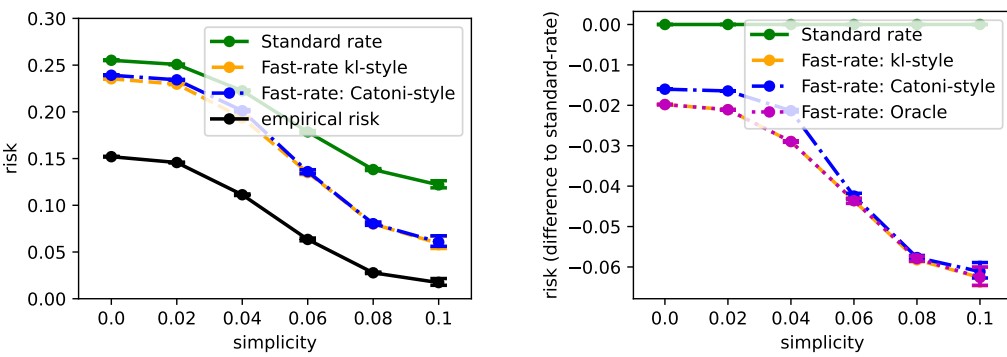

Figure 2: Sample-centric MTL: graphical results for linear models on the MDPR dataset

## 5 Experiments

While our contribution in this work is theoretical, we also provide some results from numeric experiments to illustrate the numeric behavior of the bounds we establish. Specifically, we report on experiments in two prototypical multi-task settings: linear classification with learned regularizer, and neural network learning in a random subspace representation. We also provide a visual illustration of the bounds' geometry for $n = 2$ in Appendix C.

**MTL bound for linear models.** In our first experiments, we provide generalization guarantees in the setup of linear multi-task learning with biased regularization (Kienzle & Chellapilla, 2006): for each task, we learn a (stochastic) linear classifier, parametrized by a (Gaussian) distribution over each task's classifier weights. Additionally, we learn a regularization term, which is also parametrized by a Gaussian and shared among all tasks. As dataset, we use the MDPR dataset (Pentina & Lampert, 2017), which consists of 953 tasks. Their training set sizes range between 102 and 22530 samples, making the setting clearly unbalanced. Besides the original data, we also create variants of higher *simplicity*, by adding a $\eta$-multiple of the label value to each input features for $\eta \in [0, 0.1]$. For details on the setup, see Appendix C.

Figures 1 and 2 show the results (more details and numeric values can be found in the appendix). In the task-centric setting, fast-rate and standard bounds provide similar guarantees for $\eta = 0$, but with larger values of $\eta$, the fast-rate bounds benefit more from the decrease of the empirical risk and provide tighter guarantees than the standard-rate bound. This only holds for the new unbalanced bounds we present in this work, though. The naive fast-rate bound with convergence rate determined by $m_{\min}$ is always far looser than even the standard-rate bound. In the sample-centric setting, the benefit of our fast-rate bounds is apparent already at $\eta = 0$, and the gap to the standard-rate bound increases with growing $\eta$.

Table 1: Generalization bounds for low-rank parametrized deep networks on split-CIFAR.

| Task-centric | | |
| --- | --- | --- |
| Dataset | CIFAR10 | CIFAR100 |
| Standard rate | 0.307 | 0.595 |
| Fast-rate with $m_{\min}$ | 0.347 | 0.616 |
| Fast-rate: kl-style | 0.272 | 0.591 |
| Fast-rate: Catoni-style | 0.271 | 0.591 |
| Joint constraint | 0.270 | 0.591 |
| **Sample-centric** | | |
| Dataset | CIFAR10 | CIFAR100 |
| Standard rate | 0.300 | 0.595 |
| Fast-rate: kl-style | 0.265 | 0.593 |
| Fast-rate: Catoni-style | 0.262 | 0.590 |

**MTL bound for neural networks.** Our second experiment provides generalization guarantees for unbalanced multi-task learning of low-rank parametrized neural networks (Zakerinia et al., 2025). As dataset we use *split-CIFAR10* and *split-CIFAR100* (Krizhevsky, 2009), in which the popular CIFAR datasets are split in an unbalanced way into tasks, each of which possesses only 3 of 10 (for CIFAR10) or 10 out of 100 (for CIFAR100) classes. As model, we use a Vision Transformer model with approximately $5.5$ million parameters, pretrained on ImageNet (Dosovitskiy et al., 2020). Further details are provided in Appendix C.

Table 1 shows the results. For the task-centric as well as the sample-centric views using our fast-rate bounds instead of the standard-rates ones leads to noticeable improvements, while the bound with dependence on $m_{min}$ would be much looser. In line with the theory, the improvements are larger for split-CIFAR10, where the empirical error is small, than for split-CIFAR100, where the empirical error is too large for the fast-rate property to be effective.

# 6   Conclusion

The literature on generalization behavior of multi-task learning has been heavily focused on the setting where all tasks have the same number of training samples, despite the fact that this balanced setting is not realistic for real-world tasks. In this work, we explicitly study the unbalanced multi-task learning, showing that it has different statistical properties than the balanced setting, and that the results and proof techniques from the balanced setting do not simply carry over to the unbalanced setting. We argued that for evaluating multi-task methods, there are two different views, which matter in different applications: task-centric and sample-centric. As our main contribution, we provided fast-rate generalization bounds in the PAC-Bayes framework for both settings, and we demonstrated through empirical evaluations that these bounds are not only theoretically superior but also can provide tighter guarantees than standard-rate bounds.

A limitation of our work is that the resulting bounds are less interpretable than explicit standard-rate bounds, and that numeric optimization is required to obtain numerical guarantees. We believe it will be interesting future work to explore if more explicit fast-rate bounds are possible for the unbalanced setup at all.

# Acknowledgements

This research was supported by the Scientific Service Units (SSU) of ISTA through resources provided by Scientific Computing (SciComp).

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

# A PAC-Bayesian multi-task learning

A very simple solution to achieve bounds for several tasks with different sample sizes is to compute single-task bounds for all tasks separately, and combine them by averaging and a union bound. However, this simple solution does not achieve the goal of multi-task learning, i.e., benefiting from shared information between tasks and task similarity. Importantly, this method does not share complexity terms and does not benefit from an increasing number of tasks, resulting in a sub-optimal bound, see e.g. Zakerinia et al. (2025) for numeric experiments.

PAC-Bayes multi-task learning shares information by learning a (distribution over) data-dependent prior to be shared for all tasks. Formally, we define two distributions over $\mathcal{M}(\mathcal{F}) \times \mathcal{F}^{\otimes n}$, i.e. they assign joint probabilities to tuples, $(P, f_1, ..., f_n)$, which contain a prior over models, and $n$ models: 1) $\mathfrak{Q}$ is a distribution given by the product $\mathcal{Q} \times Q_1 \times \cdots \times Q_n$, and $\mathfrak{P}$ is a distribution given by the generating process: *i)* sample a prior $P \sim \mathcal{P}$, *ii)* for each task, $i = 1, \ldots, n$, sample a model $f_i \sim P$.

Additionally, we can also define the distribution $\mathfrak{Q}$ slightly differently by the following generating process: *i)* sample a prior $P \sim \mathcal{Q}$, *ii)* for each task, $i = 1, \ldots, n$, sample a model $f_i \sim Q_i(P)$, which is a data-dependent function that adapts the posterior for each prior $P$. This approach might help reducing the complexity terms, i.e. for each $P$, the term $\mathbf{KL}(Q_i \| P)$ would become $\mathbf{KL}(Q_i(P) \| P)$. However, it causes an additional complexity in adapting the posterior to each prior. The results stated in this paper are mentioned in the form of the first approach, however, all results would hold by simply inserting the different $\mathfrak{Q}$ in the results and proofs (Pentina & Lampert, 2014; Zakerinia et al., 2024).

# B Proofs

## B.1 Auxiliary Lemmas

**Lemma B.1** ((Berend & Tassa, 2010) Proposition 3.2). *Let $X_i$, $1 \leq i \leq t$, be a sequence of independent random variables for which $P(0 \leq X_i \leq 1) = 1$, $X = \sum_{i=1}^{t} X_i$, and $\mu = E(X)$. Let $Y$ be the binomial random variable with distribution $Y \sim B\left(t, \frac{\mu}{t}\right)$. Then for any convex function $f$ we have:*

$$\mathbf{E}f(X) \leq \mathbf{E}f(Y). \tag{20}$$

**Lemma B.2** ((Maurer, 2004) Theorem 1). *Let $Y$ be the binomial random variable with distribution $Y \sim B\left(t, \frac{\mu}{t}\right)$, then we have:*

$$\mathbb{E}[e^{t\ \mathbf{kl}(\frac{Y}{t}|\frac{\mu}{t})}] \leq 2\sqrt{t} \tag{21}$$

*where $\mathbf{kl}(q|p) = q \log \frac{q}{p} + (1-q) \log \frac{1-q}{1-p}$ is the Kullback-Leibler divergence between Bernoulli distributions with mean $q$ and $p$.*

**Lemma B.3** (Generalization of (Catoni, 2007) Lemma 1.1.1.). *Let $X_{i,j} \overset{i.i.d.}{\sim} \text{Bernoulli}(\mu_i)$ for $i = 1, \ldots, n$ and $j = 1, \ldots, m_i$. Define $\hat{\mu}_i = \frac{1}{m_i} \sum_{j=1}^{m_i} X_{i,j}$, then we have:*

$$\log\left[e^{\sum_{i=1}^{n} \frac{\lambda}{n}[\Phi_{\frac{\lambda}{nm_i}}(\mu_i) - \hat{\mu}_i]}\right] \leq 1 \tag{22}$$

*where $\Phi_a(p) = -\frac{1}{a} \log(1 - p + pe^{-a})$*

*Proof.*

$$\log \mathbb{E}\left[e^{\sum_{i=1}^{n} \frac{-\lambda}{n} \hat{\mu}_i}\right] = \log \mathbb{E}\left[e^{\sum_{i=1}^{n} \frac{-\lambda}{n} \sum_{j=1}^{m_i} \frac{1}{m_i} X_{i,j}}\right] \tag{23}$$

$$= \sum_{i=1}^{n} \sum_{j=i}^{m_i} \log \mathbb{E}\left[e^{\frac{-\lambda}{nm_i} X_{i,j}}\right] = \sum_{i=1}^{n} m_i \log \mathbb{E}\left[e^{\frac{-\lambda}{nm_i} X_{i,j}}\right] \tag{24}$$

$$= \sum_{i=1}^{n} m_i \log(1 - \mu_i + \mu_i e^{\frac{\lambda}{nm_i}}) = \frac{-\lambda}{n} \sum_{i=1}^{n} \Phi_{\frac{\lambda}{nm_i}}(\mu_i) \tag{25}$$

Therefore,

$$\mathbb{E}\Big[e^{\sum_{i=1}^n \frac{-\lambda}{n}\hat{\mu}_i}\Big] = e^{\frac{-\lambda}{n}\sum_{i=1}^n \Phi_{\frac{\lambda}{nm_i}}(\mu_i)} \tag{26}$$

Multiplying both sides by $e^{\frac{\lambda}{n}\sum_{i=1}^n \Phi_{\frac{\lambda}{nm_i}}(\mu_i)}$ completes the proof. $\qquad\square$

The following lemma splits the **KL** term of (41) into more interpretable quantities.

**Lemma B.4.** *For the posterior, $\mathfrak{Q}$, and prior, $\mathfrak{P}$, defined above it holds:*

$$\mathbf{KL}(\mathfrak{Q}\|\mathfrak{P}) = \mathbf{KL}(\mathcal{Q}\|\mathcal{P}) + \mathbb{E}_{P\sim\mathcal{Q}}\Big[\sum_{i=1}^n \mathbf{KL}(A(S_i)\|P)\Big]. \tag{27}$$

*Proof.*

$$\mathbf{KL}(\mathfrak{Q}\|\mathfrak{P}) = \mathbb{E}_{P\sim\mathcal{Q}}\Big[\mathbb{E}_{(f_1,\ldots,f_n)\sim(Q_1,\ldots,Q_n)} \log \frac{\mathcal{Q}(P)\prod_{i=1}^n Q_i(f_i)}{\mathcal{P}(P)\prod_{i=1}^n P(f_i)}\Big] \tag{28}$$

$$= \mathbb{E}_{P\sim\mathcal{Q}}\Big[\log \frac{\mathcal{Q}(P)}{\mathcal{P}(P)}\Big] + \mathbb{E}_{P\sim\mathcal{Q}}\Big[\sum_{i=1}^n \mathbb{E}_{f_i\sim Q_i} \log \frac{Q_i(f_i)}{P(f_i)}\Big] \tag{29}$$

$$= \mathbf{KL}(\mathcal{Q}\|\mathcal{P}) + \mathbb{E}_{P\sim\mathcal{Q}}\sum_{i=1}^n \mathbf{KL}(Q_i\|P). \tag{30}$$

$\qquad\square$

## B.2 Task-centric bounds

**Theorem 2.3.** *For any fixed hyper-prior $\mathcal{P}$, any $\delta > 0$, it holds with probability at least $1 - \delta$ over the sampling of the training datasets, $S_1,\ldots,S_n$, that for all hyper-posterior $\mathcal{Q}$ and all posteriors $Q_1,\ldots,Q_n$:*

$$\frac{1}{n}\sum_{i=1}^n \mathcal{R}_i(Q_i) \le \frac{1}{n}\sum_{i=1}^n \widehat{\mathcal{R}}_i(Q_i) + \sqrt{\frac{\mathbf{KL}(\mathfrak{Q}\|\mathfrak{P}) + \log\frac{4m_h n}{\delta} + 1}{2m_h n}}, \tag{4}$$

*where $\mathcal{R}_i$ and $\widehat{\mathcal{R}}_i$ are the expected risk and training risk of task $i$, $m_h = \frac{n}{\sum \frac{1}{m_i}}$ is the harmonic mean of the training set sizes, $m_i = |S_i|$. $\mathfrak{Q}$ is a distribution over $\mathcal{M}(\mathcal{F}) \times \mathcal{F}^{\otimes n}$ given by the product $\mathcal{Q} \times Q_1 \times \cdots \times Q_n$, and $\mathfrak{P}$ is a distribution given by the generating process:* i) *sample a prior $P \sim \mathcal{P}$,* ii) *for each task, $i = 1,\ldots,n$, sample a model $f_i \sim P$.*

*Proof.* First for any task $i$ and any model $f_i$ we define:

$$\Delta_i(f_i) = \mathbb{E}_{z\sim D_i} \ell(z, f_i) - \frac{1}{m_i}\sum_{z\in S_i} \ell(z, f_i). \tag{31}$$

By defining $\mathcal{R}_u(\mathfrak{Q}) = \frac{1}{n}\sum_{i=1}^n \mathcal{R}_i(Q_i)$ and $\widehat{\mathcal{R}}_u(\mathfrak{Q}) = \frac{1}{n}\sum_{i=1}^n \widehat{\mathcal{R}}_i(Q_i)$, we have:

$$\mathbb{E}_{(P,f_1,\ldots,f_n)\sim\mathfrak{Q}}\Big[\frac{1}{n}\sum_{i=1}^n \Delta_i(f_i)\Big] = \mathcal{R}_u(\mathfrak{Q}) - \widehat{\mathcal{R}}_u(\mathfrak{Q}) \tag{32}$$

Applying the *change of measure inequality* (Seldin et al., 2012) between the two distributions $\mathfrak{Q}$ and $\mathfrak{P}$, for any $\lambda > 0$, and any $\mathcal{Q}, Q_1,\ldots,Q_n$, we have:

$$\mathcal{R}_u(\mathfrak{Q}) - \widehat{\mathcal{R}}_u(\mathfrak{Q}) - \frac{1}{\lambda}\mathbf{KL}(\mathfrak{Q}\|\mathfrak{P}) \le \frac{1}{\lambda}\log \mathbb{E}_{(P,f_1,f_2,\ldots,f_n)\sim\mathfrak{P}} \prod_{i=1}^n e^{\frac{\lambda}{n}\Delta_i(f_i)} \tag{33}$$

By the construction of $\mathfrak{P}$, we have

$$\underset{S_1,\ldots,S_n}{\mathbb{E}} \underset{(P,f_1,f_2,\ldots,f_n)\sim\mathfrak{P}}{\mathbb{E}} \prod_{i=1}^{n} e^{\frac{\lambda}{n}\Delta_i(f_i)} = \underset{S_1,\ldots,S_n}{\mathbb{E}} \underset{P\sim\mathcal{P}}{\mathbb{E}} \underset{f_1\sim P}{\mathbb{E}} \cdots \underset{f_n\sim P}{\mathbb{E}} \prod_{i=1}^{n} e^{\frac{\lambda}{n}\Delta_i(f_i)}, \tag{34}$$

and, because it is independent of $S_1,\ldots,S_n$, we can rewrite this as

$$= \underset{P\sim\mathcal{P}}{\mathbb{E}} \underset{S_1}{\mathbb{E}} \underset{f_1\sim P}{\mathbb{E}} e^{\frac{\lambda}{n}\Delta_1(f_1)} \cdots \underset{S_n}{\mathbb{E}} \underset{f_n\sim P}{\mathbb{E}} e^{\frac{\lambda}{n}\Delta_n(f_n)}. \tag{35}$$

$$= \underset{P\sim\mathcal{P}}{\mathbb{E}} \prod_{i=1}^{n} \underset{S_i}{\mathbb{E}} \underset{f_i\sim P}{\mathbb{E}} e^{\frac{\lambda}{n}\Delta_i(f_i)} \tag{36}$$

Each $\Delta_i(f_i)$ is a bounded random variable with support in an interval of size 1. By Hoeffding's lemma we have

$$\underset{S_i}{\mathbb{E}} \underset{f_i\sim P}{\mathbb{E}} e^{\frac{\lambda}{n}\Delta_i(f_i)} \leq e^{\frac{\lambda^2}{8n^2 m_i}}. \tag{37}$$

Therefore, by combining (36) and (37) we have:

$$\underset{S_1,\ldots,S_n}{\mathbb{E}} \underset{(P,f_1,f_2,\ldots,f_n)\sim\mathfrak{P}}{\mathbb{E}} \prod_{i=1}^{n} e^{\frac{\lambda}{n}\Delta_i(f_i)} \leq e^{\frac{\lambda^2}{8n^2}(\sum_{i=1}^{n}\frac{1}{m_i})} = e^{\frac{\lambda^2}{8nm_h}}. \tag{38}$$

Which $m_h = \frac{n}{\sum_{i=1}^{n}\frac{1}{m_i}}$ is the harmonic mean of $m_i$s. By Markov's inequality, for any $\epsilon > 0$ we have

$$\mathbb{P}_{S_1,\ldots,S_n}\left( \underset{(P,f_1,f_2,\ldots,f_n)\sim\mathfrak{P}}{\mathbb{E}} \prod_{i=1}^{n} e^{\frac{\lambda}{n}\Delta_i(f_i)} \geq e^{\epsilon} \right) \leq e^{\frac{\lambda^2}{8nm_h}-\epsilon} \tag{39}$$

Hence by combining (33) and (39) we get for any $\epsilon$:

$$\mathbb{P}_{S_1,\ldots,S_n}\left( \exists\mathfrak{Q} : \mathcal{R}_u(\mathfrak{Q}) - \widehat{\mathcal{R}}_u(\mathfrak{Q}) - \frac{1}{\lambda}\mathbf{KL}(\mathfrak{Q}\|\mathfrak{P}) \geq \frac{1}{\lambda}\epsilon \right) \leq e^{\frac{\lambda^2}{8nm_h}-\epsilon}, \tag{40}$$

or, equivalently, it holds for any $\delta > 0$ with probability at least $1 - \frac{\delta}{2}$:

$$\forall\mathfrak{Q} : \mathcal{R}_u(\mathfrak{Q}) - \widehat{\mathcal{R}}_u(\mathfrak{Q}) \leq \frac{1}{\lambda}\mathbf{KL}(\mathfrak{Q}\|\mathfrak{P}) + \frac{1}{\lambda}\log(\frac{2}{\delta}) + \frac{\lambda}{8nm_h}. \tag{41}$$

By applying a union bound for $\lambda^* \in \{1, 2, \ldots, 4nm_h\}$, and optimizing for $\lambda$ in this set we get:

$$\mathcal{R}_u(\mathfrak{Q}) - \widehat{\mathcal{R}}_u(\mathfrak{Q}) \leq \sqrt{\frac{\mathbf{KL}(\mathfrak{Q}\|\mathfrak{P}) + \log(\frac{4nm_h}{\delta}) + 1}{2nm_h}}. \tag{42}$$

$\square$

In the following theorem, we prove that standard fast-rate kl-bounds scale at best with $nm_{min}$.

**Lemma 3.1.** *Let $X_{i,j} \overset{i.i.d.}{\sim}$ Bernoulli($\mu$) for $i = 1,\ldots,n$ and $j = 1,\ldots,m_i$. Define $\widehat{\mu} = \frac{1}{n}\sum_{i=1}^{n}\frac{1}{m_i}\sum_{j=1}^{m_i} X_{i,j}$, as their average, weighted by inverse dataset sizes. Let $m_{min} = \min_i m_i$, Then, the MGF $M_\mu(\lambda) = \mathbb{E}\left[e^{n\lambda\,\mathbf{kl}(\widehat{\mu}|\mu)}\right]$ has the property that $\sup_{0<\mu<1} M_\mu(\lambda) = +\infty$, whenever $\lambda > m_{min}$. In particular, there is no finite bound on $M_\mu(\lambda)$ that depends only on $n$ and the $m_i$ but not $\mu$.*

*Proof.* Assume $\lambda > m_{min}$. Then

$$M(\lambda) = \mathbb{E}\left[e^{n\lambda\,\mathbf{kl}(\widehat{\mu}|\mu)}\right] \tag{43}$$

$$= \sum_{k_1=0}^{m_1} \cdots \sum_{k_n=0}^{m_n} \left[\prod_{i=1}^{n}\binom{m_i}{k_i}\mu^{k_i}(1-\mu)^{m_i-k_i}\right] e^{n\lambda\,\mathbf{kl}(\frac{1}{n}\sum_{i=1}^{n}\frac{k_i}{m_i}|\mu)} \tag{44}$$

$$= \sum_{k_1,\ldots,k_n} \mu^{\sum_i k_i}(1-\mu)^{\sum_i(m_i-k_i)}\left(\frac{\frac{1}{n}\sum_i\frac{k_i}{m_i}}{\mu}\right)^{\lambda\sum_i\frac{k_i}{m_i}}\left(\frac{\frac{1}{n}\sum_i\frac{m_i-k_i}{m_i}}{1-\mu}\right)^{\lambda\sum_i\frac{m_i-k_i}{m_i}}\prod_{i=1}^{n}\binom{m_i}{k_i} \tag{45}$$

$$= \sum_{k_1,\ldots,k_n} \mu^{\sum_i k_i - \lambda\sum_i\frac{k_i}{m_i}}(1-\mu)^{\sum_i(m_i-k_i)-\lambda\sum_i\frac{m_i-k_i}{m_i}} f(n,\lambda,k_1,m_1,\ldots,k_n,m_n), \tag{46}$$

where

$$f(n, \lambda, k_1, m_1, \ldots, k_n, m_n) = \left(\frac{1}{n}\sum_{i=1}^{n}\frac{k_i}{m_i}\right)^{\lambda\sum_i \frac{k_i}{m_i}}\left(\frac{1}{n}\sum_{i=1}^{n}\frac{m_i-k_i}{m_i}\right)^{\lambda\sum_i \frac{m_i-k_i}{m_i}}\prod_{i=1}^{n}\binom{m_i}{k_i}. \quad (47)$$

Assume $l$ is the index for the minimum $m_i$ such that $m_l = m_{min}$, pick the single term with $k_l = m_l$ and $k_i = 0$ for $i \neq l$. Since all terms are positive,

$$M(\lambda) \geq \mu^{m_l - \lambda\frac{m_l}{m_l}}(1-\mu)^{\sum_{i\neq l}m_i - (n-1)\lambda} f(n, \lambda, 0, m_1, \ldots, m_l, m_l, \ldots, 0, m_n) \quad (48)$$
$$(49)$$

which for $\mu < 0.1$, there is a constant $C$ such that

$$M(\lambda) > C\mu^{m_l - \lambda} f(n, \lambda, 0, m_1, \ldots, m_l, m_l, \ldots, 0, m_n). \quad (50)$$

Since $\lambda > m_l$, the exponent $m_l - \lambda < 0$, so as $\mu \to 0$, $\mu^{m_l - \lambda} \to +\infty$. Hence $M(\lambda)$ is unbounded for $\lambda > m_{\min}$, and no finite bound depending only on $n$ and the $m_i$ can hold. $\quad\square$

**Theorem B.5** (Theorem 3.2, Equation (9)). *For any fixed hyper-prior $\mathcal{P}$, any $\delta > 0$, it holds with probability at least $1 - \delta$ over the sampling of the training datasets that for all hyper-posterior functions $\mathcal{Q}$ and all posteriors $Q_1, \ldots, Q_n$ :*

$$\sum_{i=1}^{n} m_i \, \mathbf{kl}(\widehat{\mathcal{R}}_i(Q_i)|\mathcal{R}_i(Q_i)) \leq \mathbf{KL}(\mathfrak{Q}\|\mathfrak{P}) + \log\frac{1}{\delta} + \sum_{i=1}^{n}\log(2\sqrt{m_i}) \quad (51)$$

*Proof.* Based on Jensen's inequality we have:

$$\sum_{i=1}^{n} m_i \, \mathbf{kl}(\widehat{\mathcal{R}}_i(Q_i)|\mathcal{R}_i(Q_i)) \leq \mathop{\mathbb{E}}_{f_i \sim Q_i, i \in [n]}\left[\sum_{i=1}^{n} m_i \, \mathbf{kl}(\widehat{\mathcal{R}}_i(f_i)|\mathcal{R}_i(f_i))\right] \quad (52)$$

And by change of measures we get

$$\mathop{\mathbb{E}}_{P\sim\mathcal{Q}, f_i\sim Q_i, i\in[n]}\left[\sum_{i=1}^{n} m_i \, \mathbf{kl}(\widehat{\mathcal{R}}_i(f_i)|\mathcal{R}_i(f_i))\right] - \mathbf{KL}(\mathfrak{Q}\|\mathfrak{P})$$

$$\leq \log\mathop{\mathbb{E}}_{P\sim\mathcal{P}, f_i\sim P, i\in[n]}\left[e^{\sum_{i=1}^{n} m_i \, \mathbf{kl}(\widehat{\mathcal{R}}_i(f_i)|\mathcal{R}_i(f_i))}\right]$$

$$= \log\mathop{\mathbb{E}}_{P\sim\mathcal{P}}\prod_{i=1}^{n}\mathop{\mathbb{E}}_{f_i\sim P}\left[e^{m_i \, \mathbf{kl}(\widehat{\mathcal{R}}_i(f_i)|\mathcal{R}_i(f_i))}\right] \quad (53)$$

Where the equality comes from priors being data-independent. Additionally, based on Lemma (B.1), for a binomial random variable as $Y \sim B(m_i, \mathcal{R}_i(f_i))$ we have:

$$\mathop{\mathbb{E}}_{S_i\sim D_i^{m_i}}\mathop{\mathbb{E}}_{f_i\sim P}\left[e^{m_i \, \mathbf{kl}(\widehat{\mathcal{R}}_i(f_i)|\mathcal{R}_i(f_i))}\right] = \mathop{\mathbb{E}}_{f_i\sim P}\mathop{\mathbb{E}}_{S_i\sim D_i^{m_i}}\left[e^{m_i \, \mathbf{kl}(\widehat{\mathcal{R}}_i(f_i)|\mathcal{R}_i(f_i))}\right] \quad (54)$$

$$\leq \mathop{\mathbb{E}}_{f_i\sim P}\mathop{\mathbb{E}}_{Y}\left[e^{m_i \, \mathbf{kl}(\frac{Y}{m_i}|\mathcal{R}_i(f_i))}\right] \leq 2\sqrt{m_i} \quad (55)$$

Where the last inequality comes from Lemma (B.2). Therefore

$$\mathop{\mathbb{E}}_{S_1,\ldots,S_n}\mathop{\mathbb{E}}_{P\sim\mathcal{P}}\prod_{i=1}^{n}\mathop{\mathbb{E}}_{f_i\sim P}\left[e^{m_i \, \mathbf{kl}(\widehat{\mathcal{R}}_i(f_i)|\mathcal{R}_i(f_i))}\right] \leq \prod_{i=1}^{n} 2\sqrt{m_i} \quad (56)$$

And by Markov's inequality, with probability at least $1 - \delta$ over sampling of $S_1, \ldots, S_n$, we have:

$$\log\mathop{\mathbb{E}}_{P\sim\mathcal{P}}\prod_{i=1}^{n}\mathop{\mathbb{E}}_{f_i\sim P}\left[e^{m_i \, \mathbf{kl}(\widehat{\mathcal{R}}_i(f_i)|\mathcal{R}_i(f_i))}\right] \leq \sum_{i=1}^{n}\log 2\sqrt{m_i} + \log\frac{1}{\delta} \quad (57)$$

By combining (52), (53) and (57) we get

$$\sum_{i=1}^{n} m_i \, \mathbf{kl}(\widehat{\mathcal{R}}_i(Q_i)|\mathcal{R}_i(Q_i)) \leq \mathbf{KL}(\mathfrak{Q}\|\mathfrak{P}) + \log\frac{1}{\delta} + \sum_{i=1}^{n}\log(2\sqrt{m_i}) \quad (58)$$

$$\square$$

**Theorem B.6** (Theorem 3.2, Equation (10)). *For any fixed hyper-prior $\mathcal{P}$, any $\delta > 0$, and any $\lambda_1, \ldots, \lambda_n > 0$, it holds with probability at least $1 - \delta$ over the sampling of the training datasets that for all hyper-posterior functions $\mathcal{Q}$ and all posteriors $Q_1, \ldots, Q_n$:*

$$-\sum_{i=1}^{n} m_i \log(1 - \mathcal{R}_i(Q_i) + \mathcal{R}_i(Q_i)e^{\frac{-\lambda_i}{nm_i}}) \leq \frac{1}{n}\sum_{i=1}^{n} \lambda_i \widehat{\mathcal{R}}_i(Q_i) + \mathbf{KL}(\mathfrak{Q}\|\mathfrak{P}) + \log(\frac{1}{\delta}) \quad (59)$$

*Proof.* Note that the function $\Phi_a$ is convex for $a > 0$, therefore $\frac{\lambda}{n}\Phi_{\frac{\lambda}{nm_i}}$ is also convex. Based on Jensen's inequality we have:

$$\frac{\lambda}{n}[\Phi_{\frac{\lambda}{nm_i}}(\mathcal{R}_i(Q_i)) - \widehat{\mathcal{R}}_i(Q_i)] \leq \mathop{\mathbb{E}}_{f_i \sim Q_i} \frac{\lambda}{n}[\Phi_{\frac{\lambda}{nm_i}}(\mathcal{R}_i(f_i)) - \widehat{\mathcal{R}}_i(f_i)] \quad (60)$$

And by change of measures, and independence of priors, we get

$$\mathop{\mathbb{E}}_{P \sim \mathcal{Q}, f_i \sim Q_i}\Big[\frac{\lambda}{n}\sum_{i=1}^{n}[\Phi_{\frac{\lambda}{nm_i}}(\mathcal{R}_i(f_i)) - \widehat{\mathcal{R}}_i(f_i)]\Big] - \mathbf{KL}(\mathfrak{Q}\|\mathfrak{P})$$

$$\leq \log \mathop{\mathbb{E}}_{P \sim \mathcal{P}} \mathop{\mathbb{E}}_{f_i \sim P}\Big[e^{\sum_{i=1}^{n} \frac{\lambda}{n}[\Phi_{\frac{\lambda}{nm_i}}(\mathcal{R}_i(f_i)) - \widehat{\mathcal{R}}_i(f_i)]}\Big] \quad (61)$$

Now we apply Lemma (B.1) $n$ times for each task separately, keeping the others fixed. if we define a binomial random variable $Y_i \sim B(m_i, \mathcal{R}_i(f_i))$, we have

$$\mathop{\mathbb{E}}_{S_i \sim D_i^{m_i}} \mathop{\mathbb{E}}_{P \sim \mathcal{P}} \mathop{\mathbb{E}}_{f_i \sim P}\Big[e^{\sum_{i=1}^{n} \frac{\lambda}{n}[\Phi_{\frac{\lambda}{nm_i}}(\mathcal{R}_i(f_i)) - \widehat{\mathcal{R}}_i(f_i)]}\Big] \quad (62)$$

$$\mathop{\mathbb{E}}_{P \sim \mathcal{P}} \mathop{\mathbb{E}}_{f_i \sim P} \mathop{\mathbb{E}}_{S_i \sim D_i^{m_i}}\Big[e^{\sum_{i=1}^{n} \frac{\lambda}{n}[\Phi_{\frac{\lambda}{nm_i}}(\mathcal{R}_i(f_i)) - \widehat{\mathcal{R}}_i(f_i)]}\Big] \quad (63)$$

$$\leq \mathop{\mathbb{E}}_{Y_1, \ldots, Y_n}\Big[e^{\frac{\lambda}{n}[\Phi_{\frac{\lambda}{nm_i}}(\mathcal{R}_i(f_i)) - \frac{Y_i}{m_i}]}\Big] \leq 1 \quad (64)$$

Where the last inequality comes from Lemma (B.3). Therefore by Markov's inequality, with probability at least $1 - \delta$ over sampling of $S_1, \ldots, S_n$, we have:

$$\log \mathop{\mathbb{E}}_{P \sim \mathcal{P}} \mathop{\mathbb{E}}_{f_i \sim P}\Big[e^{\sum_{i=1}^{n} \frac{\lambda}{n}[\Phi_{\frac{\lambda}{nm_i}}(\mathcal{R}_i(f_i)) - \widehat{\mathcal{R}}_i(f_i)]}\Big] \leq \log\frac{1}{\delta} \quad (65)$$

By combining (60), (61) and (65) we get that with probability at least $1 - \delta$:

$$\frac{\lambda}{n}\sum_{i=1}^{n}[\Phi_{\frac{\lambda}{nm_i}}(\mathcal{R}_i(Q_i)) - \widehat{\mathcal{R}}_i(Q_i)] \leq \mathbf{KL}(\mathfrak{Q}\|\mathfrak{P}) + \log\frac{1}{\delta} \quad (66)$$

Or equivalently,

$$\frac{-1}{\lambda}\sum_{i=1}^{n} m_i \log(1 - \mathcal{R}_i(Q_i) + \mathcal{R}_i(Q_i)e^{\frac{-\lambda}{nm_i}}) \leq \widehat{\mathcal{R}}_u + \frac{\mathbf{KL}(\mathfrak{Q}\|\mathfrak{P}) + \log(\frac{1}{\delta})}{\lambda} \quad (67)$$

$$-\sum_{i=1}^{n} m_i \log(1 - \mathcal{R}_i(Q_i) + \mathcal{R}_i(Q_i)e^{\frac{-\lambda_i}{nm_i}}) \leq \frac{1}{n}\sum_{i=1}^{n} \lambda_i \widehat{\mathcal{R}}_i(Q_i) + \mathbf{KL}(\mathfrak{Q}\|\mathfrak{P}) + \log(\frac{1}{\delta}) \quad (68)$$

$\square$

## B.3 Sample-centric bounds

**Theorem B.7** (Theorem 3.3, Equation (13)). *For any fixed hyper-prior $\mathcal{P}$, any $\delta > 0$, it holds with probability at least $1 - \delta$ over the sampling of the training datasets that for all hyper-posterior functions $\mathcal{Q}$ and all posteriors $Q_1, \ldots, Q_n$:*

$$\mathrm{kl}(\widehat{\mathcal{R}}^S | \mathcal{R}^S) \leq \frac{\mathbf{KL}(\mathfrak{Q}\|\mathfrak{P}) + \log\frac{2\sqrt{M}}{\delta}}{M} \qquad \text{sample-centric } \mathbf{kl}\text{-bound} \quad (69)$$

*Proof.* Based on Jensen's inequality we have:

$$M\,\mathbf{kl}\left(\sum_{i=1}^{n}\frac{m_i\widehat{\mathcal{R}}_i(Q_i)}{M}\,\middle|\,\sum_{i=1}^{n}\frac{m_i\mathcal{R}_i(Q_i)}{M}\right) \le \mathop{\mathbb{E}}_{f_i\sim Q_i, i\in[n]}\left[M\,\mathbf{kl}\Big(\sum_{i=1}^{n}\frac{m_i\widehat{\mathcal{R}}_i(f_i)}{M}\,\middle|\,\sum_{i=1}^{n}\frac{m_i\mathcal{R}_i(f_i)}{M}\Big)\right] \tag{70}$$

And by definition of $\mathcal{R}_i$ and $\widehat{\mathcal{R}}_i$ we get

$$\mathop{\mathbb{E}}_{f_i\sim Q_i}\left[M\,\mathbf{kl}\Big(\sum_{i=1}^{n}\frac{m_i\widehat{\mathcal{R}}_i(f_i)}{M}\,\middle|\,\sum_{i=1}^{n}\frac{m_i\mathcal{R}_i(f_i)}{M}\Big)\right] = \mathop{\mathbb{E}}_{f_i\sim Q_i}\left[M\,\mathbf{kl}\Big(\frac{1}{M}\sum_{i=1}^{n}\sum_{j=1}^{m_i}\ell(f_i,z_{i,j})\,\middle|\,\mathcal{R}_w\Big)\right] \tag{71}$$

And by change of measures we get

$$\mathop{\mathbb{E}}_{P\sim\mathcal{Q}, f_i\sim Q_i}\left[M\,\mathbf{kl}\left(\frac{1}{M}\sum_{i=1}^{n}\sum_{j=1}^{m_i}\ell(f_i,z_{i,j})\,\middle|\,\mathcal{R}_w\right)\right] - \mathbf{KL}(\mathfrak{Q}\|\mathfrak{P})$$

$$\le \log\mathop{\mathbb{E}}_{P\sim\mathcal{P}, f_i\sim P}\left[e^{M\,\mathbf{kl}(\frac{1}{M}\sum_{i=1}^{n}\sum_{j=1}^{m_i}\ell(f_i,z_{i,j})\,\middle|\,\mathcal{R}_w)}\right]$$

Additionally, we have

$$\mathop{\mathbb{E}}_{z_{i,j}}\left[\frac{1}{M}\sum_{i=1}^{n}\sum_{j=1}^{m_i}\ell(f_i,z_{i,j})\right] = \mathcal{R}_w \tag{72}$$

Therefore, Based on Lemma (B.1) if we define a binomial random variable $Y\sim B(M,\mathcal{R}_w)$, we can replace the loss of samples of all tasks as in

$$\mathop{\mathbb{E}}_{S_i}\mathop{\mathbb{E}}_{f_i\sim P}\left[e^{M\,\mathbf{kl}(\frac{1}{M}\sum_{i=1}^{n}\sum_{j=1}^{m_i}\ell(f_i,z_{i,j})|\mathcal{R}_w)}\right] \le \mathop{\mathbb{E}}_{Y}\left[e^{M\,\mathbf{kl}(\frac{Y}{M}|\mathcal{R}_w)}\right] \tag{73}$$

And by Lemma (B.2) we have

$$\mathop{\mathbb{E}}_{Y}\left[e^{M\,\mathbf{kl}(\frac{Y}{M}|\mathcal{R}_w)}\right] \le 2\sqrt{M} \tag{74}$$

Therefore,

$$\mathop{\mathbb{E}}_{S_1,\ldots,S_n}\mathop{\mathbb{E}}_{P\sim\mathcal{P}}\mathop{\mathbb{E}}_{f_i\sim P}\left[e^{M\,\mathbf{kl}(\frac{1}{M}\sum_{i=1}^{n}\sum_{j=1}^{m_i}\ell(f_i,z_{i,j})|\mathcal{R}_w)}\right] \le 2\sqrt{M} \tag{75}$$

And by Markov's inequality, with probability at least $1-\delta$ over sampling of $S_1,\ldots,S_n$, we have:

$$\log\mathop{\mathbb{E}}_{P\sim\mathcal{P}}\mathop{\mathbb{E}}_{f_i\sim P}\left[e^{M\,\mathbf{kl}(\frac{1}{M}\sum_{i=1}^{n}\sum_{j=1}^{m_i}\ell(f_i,z_{i,j})|\mathcal{R}_w)}\right] \le \log\frac{2\sqrt{M}}{\delta} \tag{76}$$

By combining (70), (72) and (76) we get

$$M\,\mathbf{kl}\left(\widehat{\mathcal{R}}_w\,\middle|\,\mathcal{R}_w\right) \le \mathbf{KL}(\mathfrak{Q}\|\mathfrak{P}) + \log\frac{2\sqrt{M}}{\delta} \tag{77}$$

$\square$

**Theorem B.8** (Theorem 3.3, Equation (14)). *For any fixed hyper-prior $\mathcal{P}$, any $\delta > 0$, and any $\lambda > 0$, it holds with probability at least $1-\delta$ over the sampling of the training datasets that for all hyper-posterior functions $\mathcal{Q}$ and all posteriors $Q_1,\ldots,Q_n$*

$$\frac{-M}{\lambda}\log(1 - \mathcal{R}^S + \mathcal{R}^S e^{\frac{-\lambda}{M}}) \le \widehat{\mathcal{R}}^S + \frac{\mathbf{KL}(\mathfrak{Q}\|\mathfrak{P}) + \log(\frac{1}{\delta})}{\lambda} \quad \text{sample-centric Catoni-bound} \tag{78}$$

*Proof.* Based on Jensen's inequality we have:

$$\lambda[\Phi_{\frac{\lambda}{M}}(\sum_{i=1}^{n}\frac{m_i\mathcal{R}_i(Q_i)}{M}) - \sum_{i=1}^{n}\frac{m_i\widehat{\mathcal{R}}_i(Q_i)}{M}] \le \mathop{\mathbb{E}}_{f_i \sim Q_i} \lambda[\Phi_{\frac{\lambda}{M}}(\sum_{i=1}^{n}\frac{m_i\mathcal{R}_i(f_i)}{M}) - \sum_{i=1}^{n}\frac{m_i\widehat{\mathcal{R}}_i(f_i)}{M}]$$

(79)

And by change of measures, and independence of priors, we get

$$\mathop{\mathbb{E}}_{P \sim \mathcal{Q}, f_i \sim Q_i}\Big[\lambda[\Phi_{\frac{\lambda}{M}}(\sum_{i=1}^{n}\frac{m_i\mathcal{R}_i(f_i)}{M}) - \sum_{i=1}^{n}\frac{m_i\widehat{\mathcal{R}}_i(f_i)}{M}]\Big] - \mathbf{KL}(\mathfrak{Q}\|\mathfrak{P})$$

$$\le \log \mathop{\mathbb{E}}_{P \sim \mathcal{P}} \mathop{\mathbb{E}}_{f_i \sim P}\Big[e^{\lambda[\Phi_{\frac{\lambda}{M}}(\sum_{i=1}^{n}\frac{m_i\mathcal{R}_i(f_i)}{M}) - \sum_{i=1}^{n}\frac{m_i\widehat{\mathcal{R}}_i(f_i)}{M}]}\Big]$$

(80)

Therefore, Based on Lemma (B.1) if we define a binomial random variable $Y \sim B(M, \mathcal{R}_w)$, we can replace the loss of samples of all tasks as in

$$\mathop{\mathbb{E}}_{S_i \sim D_i^{m_i}} \mathop{\mathbb{E}}_{P \sim \mathcal{P}} \mathop{\mathbb{E}}_{f_i \sim P}\Big[e^{\lambda[\Phi_{\frac{\lambda}{M}}(\sum_{i=1}^{n}\frac{m_i\mathcal{R}_i(f_i)}{M}) - \sum_{i=1}^{n}\frac{m_i\widehat{\mathcal{R}}_i(f_i)}{M}]}\Big]$$

(81)

$$\le \mathop{\mathbb{E}}_{Y}\Big[e^{\mathbb{E}_{P \sim \mathcal{P}}\mathbb{E}_{f_i \sim P}\Big[e^{\lambda(\Phi_{\frac{\lambda}{M}}(\mathcal{R}_w) - \frac{Y}{M})}\Big]}\Big] \le 1$$

(82)

Where the last inequality comes from applying Lemma (B.3). Therefore by Markov's inequality, with probability at least $1 - \delta$ over sampling of $S_1, \ldots, S_n$, we have:

$$\log \mathop{\mathbb{E}}_{P \sim \mathcal{P}} \mathop{\mathbb{E}}_{f_i \sim P}\Big[e^{\lambda[\Phi_{\frac{\lambda}{M}}(\sum_{i=1}^{n}\frac{m_i\mathcal{R}_i(f_i)}{M}) - \sum_{i=1}^{n}\frac{m_i\widehat{\mathcal{R}}_i(f_i)}{M}]}\Big] \le \log\frac{1}{\delta}$$

(83)

By combining (79), (80) and (83) we get that with probability at least $1 - \delta$:

$$\lambda[\Phi_{\frac{\lambda}{M}}(\mathcal{R}_w) - \widehat{\mathcal{R}}_w] \le \mathbf{KL}(\mathfrak{Q}\|\mathfrak{P}) + \log\frac{1}{\delta}$$

(84)

Or equivalently,

$$\frac{-M}{\lambda}\log(1 - \mathcal{R}_w + \mathcal{R}_w e^{\frac{-\lambda}{M}}) \le \widehat{\mathcal{R}}_w + \frac{\mathbf{KL}(\mathfrak{Q}\|\mathfrak{P}) + \log(\frac{1}{\delta})}{\lambda}$$

(85)

$\square$

## B.4 Meta-Learning

In this section, we extend our results to meta-learning, i.e. learning an algorithm to use for future tasks based on unbalanced training tasks where task environments have different sample sizes in $[1, m_{max}]$, using the framework introduced in (Zakerinia et al., 2024). Similar to multi-task learning, the previous works only focused to a task-centric view, however, it is also possible to study sample-centric meta-learning: when we care more about the performance of the tasks with more sample sizes, and can define the two following meta-learning objectives:

$$\mathcal{R}_M^T(\rho) = \mathop{\mathbb{E}}_{A \sim \rho} \mathop{\mathbb{E}}_{(D,m,S) \sim \tau} \mathop{\mathbb{E}}_{z \sim D} \ell(z, A(S))$$

(86)

$$\mathcal{R}_M^S(\rho) = \mathop{\mathbb{E}}_{A \sim \rho} \mathop{\mathbb{E}}_{(D,m,S) \sim \tau} \mathop{\mathbb{E}}_{z \sim D} \frac{m}{m_{max}}\ell(z, A(S))$$

(87)

where $\rho$ is a meta-posterior over the set of algorithms $\mathcal{A}$. Each algorithm in $A \in \mathcal{A}$ is a function that given a dataset would generate a posterior distribution $A(S)$. Additionally, each algorithm can learn a hyper-posterior $\mathcal{Q}(A)$ similar to multi-task learning. Therefore, for each algorithm we would use the two distributions. 1) $\mathfrak{Q}(A)$: hyper-posterior $\mathcal{Q}(A)$ specific to algorithm $A$ (data-dependent) and task posteriors $A(S_i)$, and 2) $\mathfrak{P}(A)$: hyper-prior $\mathcal{P}(A)$ specific to algorithm $A$ (data-independent) and priors $P \sim \mathcal{P}(A)$.

For the proofs, we define the following intermediate objectives, which are multi-task risks for $A(S_1), \ldots, A(S_n)$:

$$\mathcal{R}_T(\rho) = \underset{A \sim \rho}{\mathbb{E}} \frac{1}{n} \sum_{i=1}^{n} \underset{z_i \sim D_i}{\mathbb{E}} \ell(z_i, A(S_i)) \tag{88}$$

$$\mathcal{R}_S(\rho) = \underset{A \sim \rho}{\mathbb{E}} \sum_{i=1}^{n} \frac{m_i}{M} \underset{z_i \sim D_i}{\mathbb{E}} \ell(z_i, A(S_i)) \tag{89}$$

We restate the notation introduced in the main body of the paper:

$$\mathbf{kl}_{m_1,\ldots,m_n}^{-1}\left(q_1, \ldots, q_n \Big| b\right) = \sup\left\{\frac{1}{n}\sum_{i=1}^{n} p_i \Big| \sum_{i=1}^{n} m_i \, \mathbf{kl}(q_i | p_i) \leq b, p_i \in [0,1]\right\} \tag{90}$$

and the Catoni counterpart

$$\Phi_{m_1,\ldots,m_n}^{-1}\left(q_1, \ldots, q_n \Big| b, \lambda_1, \ldots, \lambda_n\right) \tag{91}$$

$$= \sup\left\{\frac{1}{n}\sum_{i=1}^{n} p_i \Big| - \sum_{i=1}^{n} m_i \log(1 - p_i + p_i e^{\frac{-\lambda_i}{nm_i}}) \leq \frac{1}{n}\sum_{i=1}^{n} \lambda_i q_i + b, p_i \in [0,1]\right\} \tag{92}$$

### B.4.1  Task-centric meta-learning

**Theorem B.9** (Theorem 4.1, Equation (18))**.** *For any fixed meta-prior $\pi$, and fixed set of $\mathfrak{P}(A)$, any $\delta > 0$, and any $\lambda_i > 0$, it holds with probability at least $1 - \delta$ over the sampling of the training datasets that for all meta-posteriors, and hyper-posteriors $\mathcal{Q}(A)$:*

$$\mathcal{R}_M^T(\rho) \leq \mathbf{kl}_n^{-1}\left(\mathbf{kl}_{m_1,\ldots,m_n}^{-1}\left(\widehat{\mathcal{R}}_1(\rho), \ldots, \widehat{\mathcal{R}}_n(\rho)\Big| C(\rho) + c_1\right)\Big| \mathbf{KL}(\rho\|\pi) + \log\frac{4\sqrt{n}}{\delta}\right) \tag{93}$$

*where $C(\rho) = \mathbf{KL}(\rho\|\pi) + \mathbb{E}_{A\sim\rho}[\mathbf{KL}(\mathfrak{Q}(A)\|\mathfrak{P}(A))] + \log\frac{2}{\delta}, \quad c_1 = \sum_{i=1}^{n} \log(2\sqrt{m_i})$*

*Proof.* Proof of this theorem consists of two steps. 1) upper-bounding $\mathcal{R}^T(\rho), \widehat{\mathcal{R}}_1(\rho), \ldots, \widehat{\mathcal{R}}_n(\rho)$, and upper-bounding $\mathcal{R}_M^T(\rho)$ based on $\mathcal{R}^T(\rho)$. For the first step, define the following distributions:

- Sample an algorithm $A \sim \rho$, sample $(P, f_1, \ldots, f_n)$ from $\mathfrak{Q}(A)$.
- Sample an algorithm $A \sim \pi$, sample $(P, f_1, \ldots, f_n)$ from $\mathfrak{P}(A)$.

By the same proof steps as Theorem B.5 by replacing $\mathfrak{Q}$ and $\mathfrak{P}$ with these two distributions, we get

$$\underset{A\sim\rho}{\mathbb{E}}\left[\sum_{i=1}^{n} m_i \, \mathbf{kl}(\widehat{\mathcal{R}}_i(A(S_i))|\mathcal{R}_i(A(S_i)))\right] \leq C(\rho) + \log\frac{1}{\delta} + \sum_{i=1}^{n} \log(2\sqrt{m_i}) \tag{94}$$

Therefore, by Jensen's inequality

$$\sum_{i=1}^{n} m_i \, \mathbf{kl}(\widehat{\mathcal{R}}_i(\rho)|\mathcal{R}_i(\rho)) \leq C(\rho) + \log\frac{1}{\delta} + \sum_{i=1}^{n} \log(2\sqrt{m_i}) \tag{95}$$

Hence, with probability at least $1 - \frac{\delta}{2}$:

$$\mathcal{R}^T(\rho) \leq \mathbf{kl}_{m_1,\ldots,m_n}^{-1}\left(\widehat{\mathcal{R}}_1(\rho), \ldots, \widehat{\mathcal{R}}_n(\rho)\Big| C(\rho) + c_1\right) \tag{96}$$

For the second part, note that by an application of the single-task learning bound, where each sample is a task from $\tau$, we get that with probability at least $1 - \frac{\delta}{2}$:

$$\mathcal{R}_M^T(\rho) \leq \mathbf{kl}_n^{-1}\left(\mathcal{R}^T(\rho)\Big| \mathbf{KL}(\rho\|\pi) + \log\frac{4\sqrt{n}}{\delta}\right) \tag{97}$$

By combining Equations 96 and 97 with a union bound, we complete the proof. $\qquad\square$

By the same reasoning, applied to Catoni-style task-centric multi-task and single-task bounds we get the following theorem:

**Theorem B.10** (task-centric meta-learning, Catoni-style bound). *For any fixed meta-prior $\pi$, and fixed set of $\mathfrak{P}(A)$, any $\delta > 0$, any $\lambda_M > 0$, and any $\lambda_i > 0, i \in [n]$ it holds with probability at least $1 - \delta$ over the sampling of the training datasets that for all meta-posteriors, and hyper-posteriors $\mathcal{Q}(A)$:*

$$\mathcal{R}_M^T(\rho) \leq \Phi_n^{-1} \left( \Phi_{m_1,\ldots,m_n}^{-1} \left( \widehat{\mathcal{R}}_1(\rho), \ldots, \widehat{\mathcal{R}}_n(\rho) \middle| C(\rho), [\lambda_i] \right) \middle| \mathbf{KL}(\rho \| \pi) + \log \frac{2}{\delta}, \lambda_M \right) \quad (98)$$

*where $C(\rho) = \mathbf{KL}(\rho \| \pi) + \mathbb{E}_{A \sim \rho}[\mathbf{KL}(\mathfrak{Q}(A) \| \mathfrak{P}(A))] + \log \frac{2}{\delta}$*

### B.4.2 Sample-centric meta-learning

**Theorem B.11** (Theorem 4.1, Equation (19)). *For any fixed meta-prior $\pi$, and fixed set of $\mathfrak{P}(A)$, any $\delta > 0$, it holds with probability at least $1 - \delta$ over the sampling of the training datasets that for all meta-posteriors, and hyper-posteriors $\mathcal{Q}(A)$:*

$$\mathcal{R}_M^S(\rho) \leq \mathbf{kl}_n^{-1} \left( \frac{M}{nm_{max}} \mathbf{kl}_M^{-1} \left( \widehat{\mathcal{R}}^S(\rho) \middle| C(\rho) + c_2 \right) \middle| \mathbf{KL}(\rho \| \pi) + \log \frac{4\sqrt{n}}{\delta} \right) \quad (99)$$

*where $C(\rho) = \mathbf{KL}(\rho \| \pi) + \mathbb{E}_{A \sim \rho}[\mathbf{KL}(\mathfrak{Q}(A) \| \mathfrak{P}(A))] + \log \frac{2}{\delta}$, and $c_2 = \log(2\sqrt{M})$.*

*Proof.* Proof of this theorem consists of two steps. 1) upper-bounding $\mathcal{R}^S(\rho)$, based on $\widehat{\mathcal{R}}^S(\rho)$, and upper-bounding $\mathcal{R}_M^S(\rho)$ based on $\mathcal{R}^S(\rho)$. Consider the same distributions as the proof of Theorem B.11. With the same steps as the proof of Theorem B.7 using these distributions we get:

$$\mathbf{kl}(\widehat{\mathcal{R}}^S(\rho) | \mathcal{R}^S(\rho)) \leq \frac{C(\rho) + \log \frac{2\sqrt{M}}{\delta}}{M} \quad (100)$$

Therefore,

$$\frac{M}{nm_{max}} \mathcal{R}^S(\rho) \leq \frac{M}{nm_{max}} \mathbf{kl}_M^{-1} \left( \widehat{\mathcal{R}}^S(\rho) \middle| C(\rho) + c_2 \right) \quad (101)$$

and from Equations (2) we get

$$\mathcal{R}_M^S(\rho) \leq \mathbf{kl}_n^{-1} \left( \frac{M}{nm_{max}} \mathcal{R}^S(\rho) \middle| \mathbf{KL}(\rho \| \pi) + \log \frac{4\sqrt{n}}{\delta} \right) \quad (102)$$

Combination of these inequality with a union bound proves the theorem. $\qquad\square$

Analogously to above, using Catoni-style sample-centric multi-task and single-task bounds we get the following theorem:

**Theorem B.12** (sample-centric meta-learning, Catoni-style bound). *For any fixed meta-prior $\pi$, and fixed set of $\mathfrak{P}(A)$, any $\delta > 0$, any $\lambda > 0$, and any $\lambda_M > 0$ it holds with probability at least $1 - \delta$ over the sampling of the training datasets that for all meta-posteriors, and hyper-posteriors $\mathcal{Q}(A)$:*

$$\mathcal{R}_M^S(\rho) \leq \Phi_n^{-1} \left( \frac{M}{nm_{max}} \Phi_M^{-1} \left( \widehat{\mathcal{R}}^S(\rho) \middle| C(\rho), \lambda \right) \middle| \mathbf{KL}(\rho \| \pi) + \log \frac{2}{\delta}, \lambda_M \right) \quad (103)$$

*where $C(\rho) = \mathbf{KL}(\rho \| \pi) + \mathbb{E}_{A \sim \rho}[\mathbf{KL}(\mathfrak{Q}(A) \| \mathfrak{P}(A))] + \log \frac{2}{\delta}$.*

## C  Experiments

### C.1  Simulation

In this section, we provide more information and intuition on the (geometric) behavior of the bounds in Theorem 3.2. In Figure 3, we illustrate the values of the bounds for a toy example for $n = 2, m_1 = 250, m_2 = 150, \widehat{\mathcal{R}}_1 = 0.2, \widehat{\mathcal{R}}_2 = 0.2, \delta = 0.05$, and $\mathbf{KL} = 10$. The shaded areas are indicators of the areas that each constraint holds, and the goal is to maximize $\frac{1}{2}(\mathcal{R}_1 + \mathcal{R}_2)$, i.e. find a feasible point as far as possible in the diagonal direction with slope 45 degrees.

Similar to single-task learning, the **kl**-bound is a two-sided bound, which limits the deviation between $\widehat{\mathcal{R}}$ to $\mathcal{R}$ and can also give a lower-bound to the true risk. It is parameter-free and can therefore be evaluated as is. In contrast, Catoni bounds only provide an upper-bounds on the risk, which emerge as convex curves in the $(R_1, R_2)$ planes, parametrized by $n$ parameters (here: $\lambda_1, \lambda_2$), which have to be chosen a priori (in a data-independent way) for the bound to hold. The purple, orange and yellow regions in the figure illustrate different choices. For example, with $\lambda_1 = \lambda_2 = 700$ (purple line), the guarantees from the Catoni-bound are worse than the ones provided by the **kl**-bound. With $\lambda_1 = \lambda_2 = 200$ (red curve), the resulting guarantees are better, though. Since a priori good values for the $\lambda$s are not clear, this observation suggests that in practice, it is beneficial to combine the constraints from both bounds by a union bound, such that the resulting guarantees are always at least as good as the better one of them.

An interesting phenomenon emerges when doing so for $\lambda_1 = 300, \lambda_2 = 400$. The resulting bound value (yellow square) is in fact smaller than the minimum of the **kl**-bound or the corresponding Catoni-bound individually, because the different geometric shapes of the constraint sets. While in this example the difference is quite small, we find this an interesting observation that cannot occur in single-task learning (where the optimization problem is one-dimensional), and might be of independent interest.

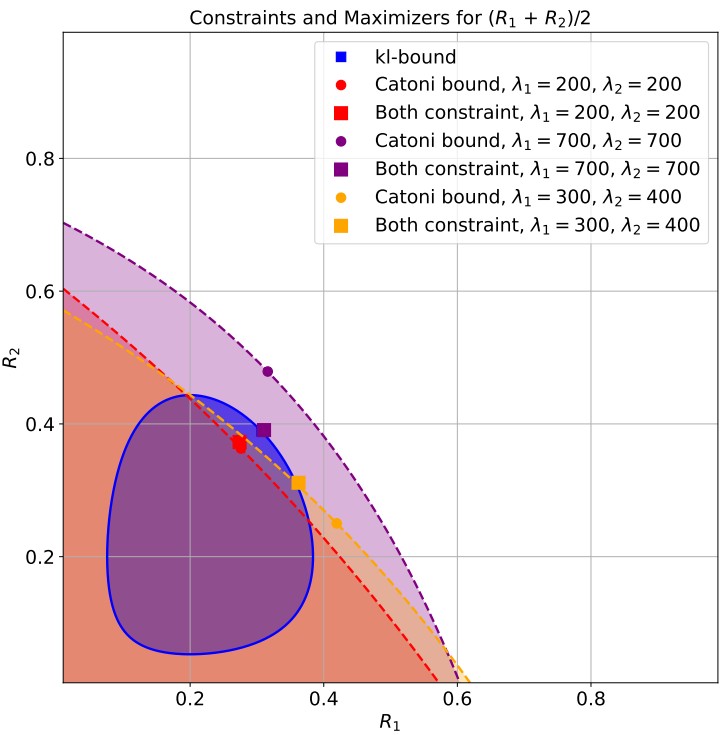

Figure 3: Illustration of the constraint and optimal value of bounds

Table 2: Task-centric MTL: numeric results (in %, lower is better) for linear models on MDPR

| simplicity | empirical risk | standard rate | fast-rate bounds | | | |
|---|---|---|---|---|---|---|
| $\eta$ | $\widehat{\mathcal{R}}^{\mathrm{T}}$ | bound | kl-style | Catoni-style | with $m_{\min}$ | oracle |
| 0.00 | $21.4_{\pm 0.1}$ | $38.8_{\pm 0.1}$ | $38.0_{\pm 0.1}$ | $38.6_{\pm 0.1}$ | $71.0_{\pm 0.1}$ | $37.4_{\pm 0.1}$ |
| 0.02 | $19.9_{\pm 0.2}$ | $37.6_{\pm 0.2}$ | $36.9_{\pm 0.2}$ | $37.3_{\pm 0.2}$ | $70.5_{\pm 0.1}$ | $36.2_{\pm 0.2}$ |
| 0.04 | $13.7_{\pm 0.1}$ | $32.5_{\pm 0.1}$ | $31.0_{\pm 0.1}$ | $30.7_{\pm 0.1}$ | $67.3_{\pm 0.2}$ | $30.4_{\pm 0.1}$ |
| 0.06 | $6.4_{\pm 0.2}$ | $25.9_{\pm 0.1}$ | $22.7_{\pm 0.1}$ | $22.6_{\pm 0.1}$ | $60.5_{\pm 0.3}$ | $22.2_{\pm 0.1}$ |
| 0.08 | $2.3_{\pm 0.1}$ | $21.0_{\pm 0.3}$ | $16.4_{\pm 0.3}$ | $16.7_{\pm 0.3}$ | $51.5_{\pm 0.7}$ | $15.8_{\pm 0.3}$ |
| 0.10 | $1.1_{\pm 0.2}$ | $18.7_{\pm 0.3}$ | $13.6_{\pm 0.3}$ | $13.6_{\pm 0.3}$ | $45.5_{\pm 0.6}$ | $13.0_{\pm 0.3}$ |

## C.2 MTL bound for linear models

In this section we provide the details for our numeric experiments on linear multi-task classification.

We use the MDPR dataset (Pentina & Lampert, 2017), which consists of 953 tasks. Each task's data is a binary classification task of 25-dimensional feature vectors, which we augment with an additional constant feature to simulate a bias term. We set aside a random subset of 500 examples from each task as the test set and use the rest for training. The resulting training set sizes range between 102 and 22530 samples (average 2450, harmonic average 859.7), making the setting clearly unbalanced. In order to influence the simplicity of the task, we create variants of the dataset in which for each task, $i$, an offset $\eta Y_i$ for $\eta \in [0, .1]$ is added to features $X_i$, where $Y_i$ is the vector of labels. Larger values of $\eta$ result in an easier classification task and consequently smaller empirical risk (and ultimately also expected risk).

As model class, we use (stochastic) linear classifiers, $f_i(x) = \mathrm{sign}(\langle x, w_i \rangle + b)$ with a Gaussian distribution of fixed variance over the weight vectors: $w_i \sim Q_i := \mathcal{N}(\mu_i, \sigma \mathrm{I}_{d \times d})$ with $\sigma = 0.1$. For training we perform logistic regression with biased Frobenius regularization (Kienzle & Chellapilla, 2006):

$$\min_{\mu_1, \ldots, \mu_n, \psi} \quad \sum_{i=1}^{n} \frac{1}{|S_i|} \sum_{(x,y) \in S_i} \mathbb{E}_{w \sim \mathcal{N}(\mu_i, \sigma \mathrm{I})} \log(1 + \exp(-y w^\top x)) + \lambda \|\mu_i - \psi\|^2 \qquad (104)$$

We implement this problem in `jax`. We determine the mean vectors $\mu_1, \ldots, \mu_n$ by 25 steps of L-BFGS optimization with $\lambda = 0.001$ using the `optax` package. To evaluate the stochastic classifiers we always sample a weight vector from the corresponding distribution. After each step, we update the regularization bias to its closed-form optimum, $\psi \leftarrow \frac{1}{n} \sum_{i=1}^{n} \mu_i$.

To evaluate the bounds, we use a hyper-prior $\mathcal{P} = \mathcal{N}(0, \mathrm{I}_{d \times d})$ and a hyper-posterior, $\mathcal{Q} = \mathcal{N}(\psi, \mathrm{I})$, such that the KL-terms become

$$\mathbf{KL}(\mathcal{Q} \| \mathcal{P}) = \frac{d}{2} \|\psi\|^2 \qquad (105)$$

$$\mathbb{E}_{P \sim \mathcal{Q}} \mathbf{KL}(Q_i \| P) = \frac{d}{2}(\sigma^2 - 1 - \log \sigma^2) + \mathbb{E}_{\psi' \sim \mathcal{N}(\psi, \mathrm{I}_{d \times d})} \frac{1}{2} \|\mu_i - \psi'\|^2 \qquad (106)$$

$$= \frac{d}{2}(\sigma^2 - \log \sigma^2) + \frac{1}{2} \|\mu_i - \psi\|^2. \qquad (107)$$

To numerically compute the non-explicit bounds, we fix $\delta = 0.05$ and first compute the empirical risks, $\widehat{\mathcal{R}}_1 \ldots, \widehat{\mathcal{R}}_n$. In the task-centric setting, we then treat the unknown risks $\mathcal{R}_1, \ldots, \mathcal{R}_n$ as free optimization variables and numerically maximize $\mathcal{R}^{\mathrm{T}} = \frac{1}{n} \sum_i \mathcal{R}_i$ subject to the inequality constraint specified to the bound of interest. In the sample-centric setting, we compute the empirical sample risk, $\widehat{\mathcal{R}}^{\mathrm{S}} = \sum_i m_i \widehat{\mathcal{R}}_i$ and solve for the scalar quantity $\mathcal{R}$, again constrained by the respective bound.

For the Catoni-style bounds, the optimal $\lambda$ value(s) are not clear a prior. Therefore, we instantiate these bounds with 11 values for $\lambda$ (where $\lambda_1 = \cdots = \lambda_n = \lambda$ in the task-centric case), exponentially spaced in $[10^{-2} \cdot nm_h, 10^2 \cdot nm_h]$ and with confidence values $\delta = \frac{0.05}{11}$. We then combine the resulting inequalities by a union bound and solve the optimization problem subject to all of the constraints. We found the `SLSQP` optimizer, which is available in `scipy.optimize`'s `minimize`'s routine to work reliably and efficiently for all of these setups.

Table 3: Sample-centric MTL: numeric results (in %, lower is better) for linear models on MDPR

| simplicity | empirical risk | standard rate | fast-rate bounds | | |
|---|---|---|---|---|---|
| $\eta$ | $\widehat{\mathcal{R}}^{\mathrm{S}}$ | bound | kl-style | Catoni-style | oracle |
| 0.00 | $15.2_{\pm 0.1}$ | $25.5_{\pm 0.1}$ | $23.5_{\pm 0.1}$ | $23.9_{\pm 0.1}$ | $23.5_{\pm 0.1}$ |
| 0.02 | $14.6_{\pm 0.1}$ | $25.1_{\pm 0.1}$ | $23.0_{\pm 0.1}$ | $23.4_{\pm 0.1}$ | $23.0_{\pm 0.1}$ |
| 0.04 | $11.1_{\pm 0.1}$ | $22.3_{\pm 0.1}$ | $19.4_{\pm 0.2}$ | $20.1_{\pm 0.2}$ | $19.4_{\pm 0.2}$ |
| 0.06 | $6.3_{\pm 0.3}$ | $17.9_{\pm 0.2}$ | $13.5_{\pm 0.3}$ | $13.6_{\pm 0.4}$ | $13.5_{\pm 0.3}$ |
| 0.08 | $2.8_{\pm 0.1}$ | $13.8_{\pm 0.2}$ | $8.0_{\pm 0.2}$ | $8.0_{\pm 0.3}$ | $8.0_{\pm 0.3}$ |
| 0.10 | $1.7_{\pm 0.6}$ | $12.2_{\pm 0.7}$ | $5.9_{\pm 1.1}$ | $6.1_{\pm 1.0}$ | $5.9_{\pm 1.1}$ |

Table 2 and 3 report the numerical results, and Figures 1 and 2 show the results graphically, which also include a plot of the differences between the new fast-rate bounds and the original standard-rate ones. Additionally included is the value of the fast-rate *oracle* bound, i.e. the Catoni-bound if the optimal $\lambda$-value(s) were known. In the task-centric case, one can see that **kl**-style and Catoni-style bounds yield comparable values, slightly (for $\eta \leq 0.2$) to clearly (for $\eta \geq 0.4$) better than the standard-rate bounds. A small difference to the oracle bound remains, indicating that a more involved procedure for choosing the $\lambda$-values might be beneficial. In the sample-centric case, the **kl**-style bound achieves almost identical results to the oracle one. The Catoni-style bound is slightly looser for $\eta \leq 0.4$, but then catches up to the other ones. Here, there is a clear improvement over the standard rate bound for all values of $\eta$.

## C.3   MTL bound for neural networks

In this section, we provide the details of the numerical experiments on multi-task learning with low-rank parametrized neural networks. [2] We use the multi-task framework of (Zakerinia et al., 2025), but generalize it to the unbalanced case. For learning $n$ models $f_1, \ldots, f_n \in \mathbb{R}^d$, they use $k$ random expansion matrices $G_1, \ldots, G_k \in \mathbb{R}^{d \times l}$, to form matrix $G = [G_1 v_1, G_2 v_2, \cdots, G_k v_k] \in \mathbb{R}^{D \times k}$, to represent a subspace, and individual models are learned in the subspace $f_j = f_0 + G \alpha_j, \alpha_j \in \mathbb{R}^k$. They follow by quantizing the model and encoding all trainable parameters using arithmetic coding.

For computing the bounds, for a subspace $G$, we would define a prior $P$ for a discrete set of models in $G$ as $P(f_j) = \frac{1}{Z} 2^{-\mathrm{len}(\alpha_j)}$, and the hyper-prior over priors (or equivalently subspaces) is $\mathcal{P}(G) = \frac{1}{Z} 2^{-\mathrm{len}(v_1, \ldots, v_k)}$. For the choice of $Q_i = \delta(f_i)$ and $\mathcal{Q} = \delta(G)$, we get $KL(\mathfrak{Q} \| \mathfrak{P}) = \mathrm{len}(G, f_1, \ldots, f_n) \log 2$, and we can compute our new bounds. For experiments, we use two multi-task benchmarks: *split-CIFAR10* and *split-CIFAR100* (Krizhevsky, 2009), which samples are distributed between tasks such that each task has data for 3 out of 10 and 10 out of 100, respectively. We use a Vision Transformer model (Dosovitskiy et al., 2020) ($\sim 5.5$ million parameters) pretrained on ImageNet. For computing the numerical upper-bound we use Sequential Least Squares Programming (SLSQP), a gradient-based optimization algorithm that solves constrained nonlinear problems by iteratively approximating them with quadratic programming subproblems. For computing the Catoni bound, we choose equal $\lambda_i = \lambda$ for all tasks. For task-centric bound we choose $\lambda = c n m_h$, and for sample-centric we choose $\lambda = cM$ for $c \in \{0.5, 0.6, \ldots, 1.5\}$ by a union bound for values of $c$. The results are given in Table 1. For both views, the improvements from slow-rates to our fast-rates are visible. Specifically, the fast-rate bounds of previous works (Guan & Lu, 2022; Zakerinia et al., 2025) for balanced MTL would give at best the rate with $m_{min}$, which translates to even a worse result compared to slow-rate bounds. Additionally, the fast-rate behavior of more improvement when training error is small is visible between the easier task (split-CIFAR10) compared to the more challenging task (split-CIFAR100).

---

[2] Code: `https://github.com/hzakerinia/MTL`

