# OpenReview forum: "Fast Rate Bounds for Multi-Task and Meta-Learning with Different Sample Sizes"
_NeurIPS.cc/2025/Conference — NeurIPS 2025 poster_

### Official Review · Reviewer_Co3j · 2025-06-21

**Clarity:** 3
**Significance:** 2
**Originality:** 2
**Rating:** 4
**Confidence:** 4

**Summary:**

The authors develop PAC-Bayesian risk bounds for the multi-task and meta-learning settings, in both cases taking care to give bounds for the unbalanced regime (where each task can have a different number of samples) that can give fast rates, even in situations where there exists a task with only a small number of samples. The focus is both on the simple average of tasks and a sample-size-weighted average (task-centric and sample-centric respectively) .The multi-task risk bounds are both of kl-type and Catoni-type, while the meta-learning risk bounds are of kl-type. Along the way, the authors show that the downside of existing ("standard") multi-task learning bounds in the unbalanced regime, while also showing how the perhaps-most-straightforward PAC-Bayesian analysis method cannot be useful when there exists a task whose number of samples is very small. The authors lightly touch upon computational issues related to getting the best possible bound from their (implicit) risk bounds. Finally, the authors demonstrate how their bounds fare in practice compared to some standard bounds and a more naive fast rate bound.

**Questions:**

1. What would you say is the core technical novelty behind your results? Having looked over some of the proofs, I think it would suffice if you restrict your attention (when answering this question) to the proof of Theorem 3.2

2. Can you point out what bound you are using when you write "Fast rate with $m\_{\\min}$" in the experimental results in the main text?

**Ethical Concerns:**

["NO or VERY MINOR ethics concerns only"]

**Final Justification:**

In my initial review, I indicated that the technical novelty behind the authors' new risk bounds is low. The author response did not provide evidence that there are new technical ideas here. I reiterate that for the proof of Theorem 3.2, the analysis is standard up until the application of Lemma A.1, and that lemma's application is similar to what happened in Zakerinia et al. (2025).

On the other hand, I agree with the authors that their results cannot be obtained in the ways suggested by one reviewer and the AC. Instead, my criticism is that what the authors actually do is a technically minor contribution compared to the proofs in previous results.

I am borderline on this paper but am not opposed to the paper being accepted.

**Limitations:**

yes

**Paper Formatting Concerns:**

All good

**Quality:**

3

**Strengths And Weaknesses:**

## Strengths

To my knowledge, prior works haven't shown fast rate PAC-Bayesian risk bounds for settings where tasks have variable sample size (the unbalanced setting). I can see that there is some value in filling in this gap, whether in the multi-task setting or the meta-learning setting. Related to the lack of prior coverage of the unbalanced setting, I don't believe the issue of bounds plagued by $m\_{\\min}$ has previously been raised, and the authors did well in highlighting how a naive approach and some existing bounds can suffer when the minimum sample size (among the training tasks) is very small.

Another interesting aspect of this work, which unfortunately was severely under-discussed in the main text, is the computational issue of going from the authors' bounds (which are implicit, involving, e.g., Bernoulli kl-divergence terms in the case of the kl-type bounds) to the best possible bound on the multi-task risk (say, the simple average). This issue strikes me as extremely interesting, and the authors apparently have thought about this a lot due to the extra material in the appendix (which, out of respect of the page limit and since it was new content not supporting results appearing in the main text, I regret to say I could not review out of fairness to the other papers in submission to NeurIPS this year). Going forward, I encourage the authors to devote significant attention to this computational aspect in the main text, including at least one figure. This might be where the key technical contribution actually lies for your work. You might even have new theorems as part of this (like, extracting the best possible explicit bound given your existing implicit bounds). In truth, the under-coverage of the computational aspect in the main text is a weakness that can be turned into a major strength.

I think it is nice that the authors made explicit the task-centric and sample-centric objectives. I can't say that this is terribly novel, as task-centric is standard and (without even checking) I feel that sample-centric should have appeared in some works before. Even so, formalizing all of this, along with presenting bounds for each case, is nice.

The experimental results are encouraging. I have a question about precisely what bound was used for "Fast rate with $m\_{\\min}$".


## Weaknesses

I felt the paper is quite lacking when it comes to describing how the technical ideas in the present paper relate to previous works. For example, Theorem 2.1 (which I understand the authors do not claim to be a novel result but simply present as a way to provide context for their upcoming, new results) seems to essentially be due to Pentina and Lampert (2014). Yet, the proof in the appendix is written from scratch, without giving credit to previous, related results.

I found to be underwhelming the technical novelty behind the authors' new risk bounds (e.g., Theorem 3.2). Inspection of the proof reveals that the authors follow a standard PAC-Bayesian analysis modulo the application of Lemma A.1; the application of that lemma is similar to how the lemma is used in the work of Zakerinia et al. (2025). I found the lack of a majorly new technical idea to be surprising because the authors say — both in their Introduction and their Section 3 — that their contribution is technical. For a venue like NeurIPS and a paper that positions it as theory paper (as the authors do), please take care to explain what the technical hurdles were.

Less major 1: From the writing, I felt the authors were suggesting that Lotfi et al. (2024) considered PAC-Bayesian bounds. I believe this work only considered compression bounds. It would be better to adjust the writing to avoid sending an incorrect message (please correct me if I'm wrong; I could not find PAC-Bayes bounds in the work of Lotfi et al.).

Less major 2: A somewhat minor presentational aspect is this the authors do not make clear where the proofs of various results happen. For example, a search for "Theorem 3.2" in the appendix does not turn up any results. Instead, I eventually found two theorems in the appendix which together provide the two statements of Theorem 3.2.


## Additional comments


A few comments on the appendix:

Line 495: $\\mathcal{R}\_u$ and $\\hat{\\mathcal{R}}\_u$ were never defined, although I could infer their meaning.

Line 506: for completeness you should mention that you're using a geometric grid rather than using a union bound over the infinite set $(1, 4 n m\_h]$.

Equation (51): Here and in some places below here, the summation should be pulled to the outside; as written, $f\_i$ is out of scope (since $i$ is "out of scope" in the computer-science sense).

Lines 527–528: It would help to provide more explanation on precisely which function $f$ you consider (when applying Lemma A.1), what choice of $X\_i$ you use, etc. I figured this out myself but it took a bit of time as I had to write everything out. Also, note that you are missing an expectation with respect to $f\_i$ just after the inequality in equation (53).

---

> ### Author Rebuttal · Authors · 2025-07-30
>
> Thank you for your helpful review. We hope our response answers your questions and addresses your concerns.
>
> > Question 1: What would you say is the core technical novelty behind [...] the proof of Theorem 3.2
>
> As common in PAC-Bayesian theory, it is not that one specific step of the proof is of high technical depth, but our contribution lies in the choice which individual objects to define and how to combine them in order to achieve optimal rates.
> For Theorem 3.2, we take insight from our lower bound in Lemma 3.1 (which is another technical contribution) and define separate terms for different tasks; however, we then use a shared change of measure step to keep as much information as possible shared among tasks and establish the optimal dependency on the sample sizes. We also bound the moment generating functions of each task separately, because the proof of Lemma 3.1 suggests that a joint bound would lead to a suboptimal rate.
>
> > Question 2: what bound you are using when you write "Fast rate with “m_min" in the experimental results in the main text?
>
> We will make this more explicit in the manuscript. It is the bound that we get by following the standard MGF bound steps of Lemma 3.1 with choice $\lambda = m_{min}$ (i.e. similar to the bounds of  Guan & Lu (2022) and Zakerinia et al. (2025).)
>
> We would also like to comment on some other parts of the review, hoping that this will address some of your other concerns.
>
> > Computational issue of going from the implicit to explicit bounds:
>
>
> Thank you for this encouraging discussion and suggestion. Indeed, we are aware of this aspect and its importance for connecting MTL theory to real applications. However, we found it orthogonal to the main results in the manuscript, which is why we included the material only in the appendix. Given your encouragement, we will try to make the aspect more prominent.
> Of course, we respect your choice not to review it. However, we would like to emphasize that the appendix was submitted at the same time as the main body of the paper, so its content is not the result of additional time, and we do refer to the appendix in the main body.
>
>
> > relation of Theorem 2.1 to previous work
>
> As you mention, Theorem 2.1 is part of the background section, not an actually new theorem. It is indeed similar to the classical results of Pentina and Lampert (2014), but the complexity term has a different (simpler and more modern) shape. We are not aware that this bound with square root dependence and different sample sizes was ever presented in prior works, so we provide its proof for completeness.  We will be happy to clarify this in the manuscript and expand the discussion.
>
> > nature of technical contribution
>
>
> We consider our contribution technical as well as conceptual (lines 42 and 148). The technical aspect lies in formally deriving the roadblock in why prior approaches were not able to derive fast rates in the unbalanced setting and in deriving a construction that overcomes them. Please also see our response above. Our fast-rate bounds could not have been derived by just following the steps of Guan&Lu (2022) or Zakerinia et al. (2025). At best a result with $\lambda= m_{min}$ like Lemma 3.1 would have emerged.
> The conceptual lies observing that two MTL scenarios make sense in the unbalanced setting, both should be treated by different bounds.
>
> > description of Lotfi et al. (2024)
>
> Thanks for noticing this, we will clarify this in the manuscript. Lotfi et al. (2022) is indeed formulated as PAC-Bayesian, and it introduces the case of a finite hypothesis set as a special case. Lotfi et al. (2024) adopts the same finite-hypothesis setup, but does not phrase it as PAC-Bayesian anymore.
>
> > connection of proof in the Appendix to theorem
>
> Thank you for the helpful feedback. We will make the connections between main text and the Appendix clearer, including explicit statements about the proofs.
>
> >Additional comments
>
> Thanks, we'll fix these.

---

> > ### Comment · Reviewer_Co3j · 2025-08-02
> > **response**
> >
> > Thanks for your response. I continue to believe that the technical contribution is moderate at best. As such, I maintain my previous evaluation. I'll keep up with the authors' exchanges with the other reviewers.

---

### Official Review · Reviewer_7kbH · 2025-07-02

**Clarity:** 4
**Significance:** 3
**Originality:** 2
**Rating:** 5
**Confidence:** 4

**Summary:**

This paper derives new, fast rates for the multi-task and meta learning problems under imbalanced sample sizes per task. The authors provide results in a PAC-Bayes framework, which situates their contributions within both the MTL/ML and PAC-Bayes literatures. Their results bring to light new phenomena in the behavior of multi-task and meta-learning with imbalances sample sizes.

**Questions:**

1. What are the key technical challenges of the proofs of Theorems 3.2, 3.3, and 4.1? What differentiates the proof techniques used in this paper from existing techniques that are, seemingly, provably suboptimal?

2. My understanding is that the novel behavior that the authors uncover in imbalanced multi-task learning is most prevalent in the task-centric setting. What is the authors' intuition for why task-centric multi-task learning is more affected by imbalanced sample sizes than the sample-centric analog?

**Ethical Concerns:**

["NO or VERY MINOR ethics concerns only"]

**Final Justification:**

The paper is well written, and the authors have answered by questions in the rebuttal regarding the technical difficulty of the paper.

**Limitations:**

Yes

**Paper Formatting Concerns:**

None.

**Quality:**

3

**Strengths And Weaknesses:**

Strengths: The paper is exceptionally well-written and clearly motivated. The authors provide an excellent background for the problem under consideration. In particular, Lemma 3.1 demonstrates formally the necessity of the contribution of this work. The main theoretical results are clearly written and genuinely novel. The simulations support the bounds numerically.

Weaknesses: It is not clear that proofs of the theorems are significantly different from existing results on a technical level. I am not an expert in PAC-Bayes, but my reading of the proofs is that they broadly follow the same structure as existing proof techniques. Upon my current reading of the paper, this slightly detracts from the work's novelty, but I look forward to discussing the nuances of the proof with the authors in the discussion section.

---

> ### Author Rebuttal · Authors · 2025-07-30
>
> Thank you for the insightful review. I hope our reply will address your questions.
>
>
> > key technical challenges of the proofs:
>
>
> Indeed, the proofs of PAC-Bayesian results often consist of similar steps, but the details matter in order to obtain tight guarantees. For Theorem 3.2, **the technical novelty lies in the choice which individual objects we define and how we combine them**: inspired by our lower bound in Lemma 3.1, we define separate terms for different tasks; however, we then use a shared change of measure step to keep as much information as possible shared among tasks and establish the optimal dependency on $n$. We also bound the moment generating functions of each task separately, because the proof of Lemma 3.1 suggests that a joint bound would lead to a suboptimal rate.
> For Theorems 3.3 and 4.1, our contribution is mostly conceptual as we introduce a new (sample-centric) objective that better matches the requirements of some multi-task applications; however, the community did not look at it before. The proof of the results only required standard existing proof techniques.
>
>
> >  What is the authors' intuition for why task-centric multi-task learning is more affected by imbalanced sample sizes than the sample-centric analog?
>
>
> The main point is that in the task-centric setting, samples from different tasks have different importance/weights, i.e. the “rare” samples from low-budget tasks influence the loss more than samples from the tasks with more samples. Controlling this difference introduces more challenges compared to the sample-centric in which all samples have the same importance.

---

> > ### Comment · Reviewer_7kbH · 2025-08-04
> >
> > Thank you for your response to my question. I suggest adding a remark in the paper clarifying the technical difficulty of your proof and how it is distinguished from existing results in the literature. Seeing as my questions were answered to satisfaction, I will increase my score accordingly.

---

### Official Review · Reviewer_cSyd · 2025-07-06

**Clarity:** 3
**Significance:** 2
**Originality:** 1
**Rating:** 4
**Confidence:** 2

**Summary:**

This paper provides fast risk bounds for multi-task learning and meta-learning, where different tasks could have different sample sizes. It argues that previous bounds cannot be easily generalized to varying sample cases, and argues for a different proof technique required for handling different sample sizes. It also evaluates the bounds numerically on various tasks and show that it improves over the previous bounds.

**Questions:**

1. Why can't the proof of Guan & Lu 2022 be generalized to different sample sizes? As stated before, I think in Proposition 6, the error can be split over various tasks separately.

**Ethical Concerns:**

["NO or VERY MINOR ethics concerns only"]

**Final Justification:**

The rebuttal response clarified the concern I had. I have updated my score accordingly.

**Limitations:**

I think the comparison to previous works and why those cannot be extended to different sample sizes needs to be properly addressed.

**Quality:**

2

**Strengths And Weaknesses:**

The main strength of the paper is the generalization of prior bounds on multi-task and meta-learning. It argues that previous bounds cannot be easily extended to the case of different sample sizes. However, in my opinion, this does not seem true. Looking at bounds of Guan & Lu 2022, the fast bounds for meta learning can be easily generalized to different sample sizes. Using their notation, $\tilde{err}(Q) - \hat{err}(Q)$ in the paper is currently bounded in Proposition 6 in the Appendix. It gives a rate of $1/mn$, assuming n tasks and m samples per task. Instead simply splitting $\tilde{err}(Q) - \hat{err}(Q)$ over different tasks (as in Proposition 3) would give the error rate dependent on each $m_i$ separately.

---

> ### Author Rebuttal · Authors · 2025-07-30
>
> Thank you for your review. Your critique seems to be based on the impression that our result lacks relevance, as you suggest an alternative technique for bounding the multi-task risk. However, we believe this to be a misunderstanding that we would like to clarify in the following.
>
>
> If we read your description right, you suggest a two-stage strategy: first, bound the risks for each task separately, and then combine the resulting bounds. The first step would indeed result in $n$ individual bounds with rates $1/m_i$. The problem emerges in the second step of how to combine the bounds into an overall multi-task risk. Specifically, the average $1/n \sum_i 1/{m_i}$ does not improve with increasing $n$. Furthermore, combining high probability bounds will typically require a union bound across tasks, which results in an additional $log(n)$ term. For example, in the balanced case, the resulting bound would have a rate $O(log(n)/m)$, which increases with the number of tasks instead of decreasing. In comparison, our bounds scale like $O(log(n)/nm)$ in this case.
>
>
> We hope that in light of these clarifications you reassess your assessment of our contribution. If you have further questions or concerns, we would be happy to answer them during the discussion phase.

---

> > ### Comment · Area_Chair_pWMu · 2025-08-08
> > **Follow-up to the comments by reviewers cSyd and 4gzb**
> >
> > I wanted to follow up on the comments from reviewers cSyd and 4gzb.
> >
> >
> > In a standard PAC-type argument, one can avoid an $m_{\min}$ bottleneck in the unbalanced setting by bounding each task’s generalization gap separately (via Bernstein or empirical Bernstein), obtaining a deviation term $O(1/m_i)$ in the fast-rate case (or $O(1/\sqrt{m_i})$ for slow rates), and then averaging these bounds with appropriate weights. The leading variance term then depends on $\sum_i 1/m_i$ (harmonic mean $m_h$) or the total sample size $M$, not $m_{\min}$. This yields rates $1/(n m_h)$ or $1/M$, so large-$m_i$ tasks converge quickly even if some $m_i$ are small.
> >
> >
> > In the PAC-Bayes setting, the proof is more delicate: the bound must hold \emph{uniformly} over all posteriors, requiring a change-of-measure step and an MGF bound under the prior. In the balanced case, one can pick a single multiplier $\lambda$ tied to $m$, but in the unbalanced case, the MGF blows up for $\lambda > m_{\min}$ (Lemma 3.1) because each task scales differently with $m_i$. Avoiding this requires per-task MGFs and combining them carefully, or bounding a weighted sum of per-task KL terms (Theorem 3.2) instead of the KL on the averaged risk, making fast-rate PAC-Bayes bounds substantially more involved than their classical PAC counterparts.
> >
> >
> > In light of the discussion above, it is concerning that the authors do not review the literature on PAC-style generalization bounds for MTRL, especially since prior work has already given fast rates in the PAC setting for the balanced case, and a simple application of the ideas above yields the result for the unbalanced case. The result of (Yousefi et al., 2016; Theorem 1) depends on $m_{\min}$ because their proof requires upper bounds with identical sample-size dependence for all tasks before combining terms, effectively forcing the worst-case $m_{\min}$ into the rate. A more refined argument that bounds each task’s deviation separately, as sketched above, avoids this bottleneck and yields dependence on $\sum_i 1/m_i$ instead.

---

> > > ### Author Response · Authors · 2025-08-08
> > >
> > > Dear AC,
> > >
> > > Thanks for engaging with us. Given the short time before the deadline, please forgive our brief style of reply.
> > >
> > > We approach the problem in a PAC-Bayesian framework in light of the fact that such bounds can be made quite tight, even for deep learning settings. We are aware of the PAC works on MTL, and we’ll be happy to include a discussion on fast-rate PAC works on MTL, but we see our contribution orthogonal to it, not building on top of it, and the problem has not been solved by PAC techniques, either, so far.
> > >
> > > About the approach you describe: it seems to resemble the strategy initially also suggested by Reviewer cSyd: one first bounds each task gap separately, and then combines the bounds. For this to result in a bound on the task-centric risk, presumably the combination would require a uniformly weighted average (you mention *appropriate weights*, so if you mean something other than uniform, we’d be thankful for detail). The complexity terms ($C_i/m_i$ for task $i$) would also be averaged, i.e. become $\frac{1}{n}\sum_i \frac{C_i}{m_i}$. Assuming comparable $C_i$’s, and inserting  $m_h=\frac{n}{\sum_i \frac{1}{m_i}}$, that’s an overall rate of $\frac{C}{m_h}$, not $\frac{C}{n m_h}$. In particular, it does not improve with the number of tasks, even in the balanced setting.
> > >
> > > Note that this makes sense. **A two-step procedure does not share any information between tasks, so no speedup can be expected when the number of tasks increases**.
> > >
> > > The crucial part of a multi-task bound is to incorporate the possibility that shared information leads to a reduction in uncertainty.
> > >
> > > For example consider the setting of representation learning that there is a shared feature extractor and task-specific heads. In the approach of bounding the tasks separately, the shared feature extractor would cost $n$ times in the complexity terms, although it is shared between all tasks.
> > >
> > > In order to gain from information sharing, we need to combine tasks at an earlier step, similar to Yousefi et al. (2016), which for them resulted in a bound with $\frac{1}{nm_{min}}$ complexity. Theirs is true multi-task bound with complexity terms that do improve with the number of tasks. For balanced task sizes the achieved bound is of optimal rate, but for unbalanced task sizes it is not.
> > >
> > > Thank you,
> > >
> > > The Authors

---

> ### Comment · Area_Chair_pWMu · 2025-08-08
>
> As I said, my question was prompted by the comment from reviewer cSyd, and I did read your response to the reviewer carefully. Here is how you could use empirical Bernstein:
>
> Apply a per-task empirical Bernstein (Audibert--Munoz--Szepesv\'ari--style) bound:
> for each $i$, with probability $\ge 1-\delta_i$,
>
> $R_i - \hat R_i \le \sqrt{\frac{2 \hat V_i \log(2/\delta_i)}{m_i}} + \frac{7 \log(2/\delta_i)}{3(m_i-1)},$
>
>
> where $\hat V_i$ is the empirical variance of losses in task $i$.
>
>
> Pick $\delta_i = \delta/n$ and take a union-bound over $i$. Averaging over tasks yields, with probability $\ge 1-\delta$,
>
>
>
> $R_T - \hat R_T \le \frac{1}{n}\sum_{i=1}^n \sqrt{\frac{2\hat V_i \log(2n/\delta)}{m_i}}+ \frac{7 \log(2n/\delta)}{3n}\sum_{i=1}^n \frac{1}{m_i-1}.$
>
>
>
> If you want a single compact rate, use $\hat V_i\le 1/4$ to get
>
> $R_T - \hat R_T \lesssim \sqrt{\frac{\log(n/\delta)}{2n}\sum_i\frac{1}{m_i}} + O\left(\frac{\log(n/\delta)}{n}\sum_i\frac{1}{m_i}\right),$
>
> again driven by $\sum_i 1/m_i$ (or $m_h$).
>
> Equal $\delta_i$ is clean, but one can sometimes reduce constants by giving
> larger $\delta_i$ (looser confidence) to tasks whose bound is already tight
> (large $m_i$) and {smaller} $\delta_i$ (tighter confidence) to tasks with few samples. For example, $\delta_i = \delta \frac{m_i^{-1}}{\sum_{k=1}^n m_k^{-1}}$ or $\delta_i = \delta \frac{m_i}{\sum_{k=1}^n m_k}.$

---

> > ### Author Response · Authors · 2025-08-08
> >
> > Dear AC, Thanks for your reply.
> >
> > Note that there is a small mistake in the provided proof.
> > In the step where you moved the average inside the square root, there should be $n$ instead of $n^2$ in the denominator.
> >
> > Based on Jensen's or Cauchy–Schwarz inequality, we would get (assuming $\hat{V}_i \le \frac{1}{4}$)
> >
> >  $\frac{1}{n}\sum_{i=1}^n \sqrt{\frac{\log(2n/\delta)}{2m_i}} \le \sqrt{\frac{\log(2n/\delta)}{2n}\sum_{i=1}^n \frac{1}{m_i}}$
> >
> > Therefore, the rate of the correct bound would be based on $\frac{1}{n} \sum_i \frac{1}{m_i} = \frac{1}{m_h}$, and not the optimal $\frac{1}{n m_h}$. This means that the bound would not improve if we get more tasks, which is not optimal.

---

> > > ### Comment · Area_Chair_pWMu · 2025-08-09
> > > **An alternate argument**
> > >
> > > This bound is not different from what you have if you write out the KL divergence between hyperpriors and hyperposteriors as a sum of task-specific KL divergences, i.e., $KL(\mathfrak{P}||\mathfrak{Q}) = \sum_i KL(P_i || Q_i).$ If we bound each of these divergences by a constant (as we did with the empirical variance), we end up with the same dependence.
> > >
> > > For the PAC setting, we can also try getting a slightly different bound. For instance, we can switch the inequality at the averaging step, i.e., instead of Jensen's inequality on the sum of square roots, we can treat the $n$ per-task gaps as independent bounded variables and apply Bernstein's (or Hoeffding's) inequality at the task level. That gives a slightly different dependence.
> > >
> > >
> > > Here’s the short derivation.
> > >
> > >
> > > Let  $X_i := R_i - \hat R_i,  i=1,\dots,n.$
> > >
> > >
> > > Then $\mathbb{E}[X_i]=0$ (since $\mathbb{E}[\hat R_i]=R_i$), and the $X_i$ are independent across tasks.
> > >
> > >
> > > Also $X_i\in[-1,1]$ because $R_i,\hat R_i\in[0,1]$. Moreover,
> > >
> > > $\operatorname{Var}(X_i) = \operatorname{Var}(\hat R_i) \le \frac{V_i}{m_i} \le \frac{1}{4m_i},$
> > > where $V_i=\operatorname{Var}[\ell(f_i,Z)]\le \tfrac14$ for losses in $[0,1]$.
> > >
> > > We want a tail bound for the task-centric average:
> > >
> > >
> > > $\Delta_T := R_T - \hat R_T = \frac{1}{n}\sum_{i=1}^n X_i.$
> > >
> > >
> > > Apply Bernstein to the independent, zero--mean, bounded $X_i/n$. The variance proxy is
> > > $\sigma^2 = \sum_{i=1}^n \operatorname{Var}\left(\frac{X_i}{n}\right) \le \sum_{i=1}^n \frac{1}{n^2}\cdot \frac{1}{4m_i} = \frac{1}{4n^2}\sum_{i=1}^n \frac{1}{m_i} = \frac{1}{4n m_h},$
> > >
> > > since $\sum_i \frac{1}{m_i} = \frac{n}{m_h}$. The range parameter is $b = \max_i |X_i|/n \le 1/n$.
> > >
> > > Bernstein yields, with probability at least $1-\delta$,
> > >
> > >
> > > $R_T - \hat R_T \le \sqrt{2 \sigma^2 \log(1/\delta)} + \frac{2}{3} b \log(1/\delta) \le  \sqrt{\frac{\log(1/\delta)}{2 n m_h}} + \frac{\log(1/\delta)}{3 n}.$
> > >
> > >
> > > If you want a variance-adaptive version, you can replace $1/(4m_i)$ above by an estimate of $\operatorname{Var}(\hat R_i)$ (e.g., $\hat V_i/m_i$), but then you need a small extra step to control the randomness of those plugins.
> > >
> > >
> > > For the sample-centric case, it is even more straightforward to get the following bound
> > >
> > > $\Delta_T  \le \sqrt{\frac{\log(1/\delta)}{2M}}+\frac{\log(1/\delta)}{3M},$
> > >
> > > where $M = \sum_i m_i$ is the total sample size and there is no $m_{\min}$ dependence.
> > >
> > >
> > > Finally, I would like to remark that all of the arguments above are rather simple and straightforward, and that they yield the same rates you obtain (and some slightly different)  for unbalanced data. I also find the bounds on excess risk more interpretable, computable, and easy to understand.

---

> > > > ### Author Response · Authors · 2025-08-09
> > > >
> > > > Dear AC, Thank you for your reply.
> > > >
> > > > > KL divergence and dependency of our bound to $n$
> > > >
> > > > In the expression for $KL(\mathfrak{Q}||\mathfrak{P})$ that you write you left out the term for the hyper-posterior $KL(\mathcal{Q}||\mathcal{P})$, which can capture shared information and --contrary to the per task terms-- is not a sum over $n$ terms. Its presence is what makes our result an actual multi-task bound.
> > > >
> > > > Let us discuss two examples that illustrate the extremes of what can happen in MTL:
> > > >
> > > > 1) If we get $n$ totally independent tasks with nothing to share, we cannot make use of the hyper-posterior term. Then, the complexity term reduces to the sum of  $n$ separate terms, as you write, and the bound won’t benefit from more tasks. This is expected, since there is no shared information between tasks.
> > > >
> > > > However, if tasks are related, our result *does* allow for information sharing by means of the hyper-posterior, thereby improving the rate.
> > > >
> > > > 2) Imagine that all tasks have actually the same distribution. Then, the complexity can be captured by the KL of hyper-posterior and hyper-prior whereas the individual terms vanish. On the right hand side of Theorem 3.2 (Eq. 8), the KL-term would be a constant, regardless of $n$, where the left hand side would still be a sum over all tasks and their samples, i.e. scale with the total sample size. Again, we believe this is natural: if all tasks have identical distributions, the best strategy is to pool their samples and learn identical models.
> > > >
> > > > Clearly, the approach of averaging separate terms works in Example 1). In example 2), that approach would still have the average of the same complexity and would not improve with $n$. Intuitively, after the concentration of each task has been quantified by some complexity measure, the exact task distributions are not “available” to the proof steps anymore, so the two extreme examples cannot be distinguished.
> > > >
> > > > This is the main reason to apply the bound jointly, to capture the sharing information between different tasks. Note that (Zakerinia et al. 2025), which our neural network experiment is based on, shows that in practice, the total complexity term (there based on the total encoding length) grows sub-linearly based on $n$, and our bound improves with $n$.
> > > >
> > > > >  The new bound
> > > >
> > > > **Note that the new bound also does not have the optimal rate. The fast-rate part of this bound does improve with $n$, however does not improve with $m_h$ anymore, for an overall rate of $O(\frac{1}{n} + \sqrt{\frac{1}{n m_h}})$ instead of $O(\frac{1}{nm_h})$.**
> > > >
> > > >
> > > > > Finally, I would like to remark that all of the arguments above are rather simple and straightforward, and that they yield the same rates you obtain (and some slightly different) for unbalanced data. I also find the bounds on excess risk more interpretable, computable, and easy to understand.
> > > >
> > > >
> > > > First, we would like to note that while concentration arguments are typically elementary, a full generalization bound, in which learning creates dependence between terms will still require more technical machinery, leading to a longer and more involved proof.
> > > >
> > > > Regardless, we would like to return the discussion to the merits of our submission. We understand that you as AC believe that easier PAC-based alternatives exist. Even though none of the constructions you provide do achieve this, we also cannot rule out that such a construction would be possible, of course. Nevertheless, I hope we agree that there is no such result in the literature (which we would find surprising, if the construction were indeed elementary), and the related published work (Yousefi et al 2016) exhibits the same limitation of a sub-optimal dependence in the unbalanced case that had triggered our analysis.
> > > >
> > > > In our submission, we adopt a different framework and we state a different kind of solution that we believe is correct, novel and improves substantially over the previously published work. We hope the decision on our work will be based on these characteristics.

---

> > > > > ### Comment · Area_Chair_pWMu · 2025-08-09
> > > > >
> > > > > The point was to show that you can avoid dependence on \min_i m_i. You said that the PAC bounds I sketched are not "optimal", but the rates in your bounds have the same behavior if we do not utilize the shared task information to derive these bounds. Of course, much work in this area has focused on understanding this task-relatedness via a notion of task diversity; all of this discussion is missing in your paper and hidden within the KL divergence between hyper-prior and hyper-posterior.
> > > > >
> > > > > I am also not sure what is described in the Experiments section, especially in Table 1. The idea of *simplicity* makes no sense in Figure 1 even after reading the Appendix B carefully.

---

> ### Author Response · Authors · 2025-08-09
>
> Dear AC,
>
> Given that there are just a few minutes left, we only provide a very short answer.
>
> We can not find any work on MTL that does not have information sharing, and our claims and analysis are also about real MTL bounds.
>
> About the rate, we wanted to improve dependency over $min_i  m_i$ to achieve a bound with a better sample complexity than $n m_{min}$. The bound you provide has the sample complexity of $n$.

---

### Official Review · Reviewer_4gzb · 2025-07-08

**Clarity:** 3
**Significance:** 3
**Originality:** 2
**Rating:** 4
**Confidence:** 4

**Summary:**

The work derives PAC-Bayes fast rates for multi-task learning (MTL) with unbalanced datasets.
There is prior work on PAC-Bayes for _balanced_ datasets, yet the authors provide a lemma that shows that the usual line of argument fails to go through in the unbalanced setting.
Thus, the authors provide an alternative argument that provides both kl-style and Catoni-style bounds.
The prior bounds are both provided in a task-centric (each task is given equal weight) and sample-centric (weight per task is proportional to number of samples for that task) form.
Previous fast-rates in the balanced MTL setting are able to be "numerically inverted" to provide bounds on the risk.
This work studies the non-trivial solution space of such numerical procedures for task-centric MTL.  Finally they conduct experiments to on linear and neural networks that demonstrate the benefit of their approach versus the rate that doesn't scale in aggregate (but with the size of the minimum sample) and the standard rate.

**Questions:**

* Is my remark about the meta-learning proofs above representative?

* Reviewing the proof of Lemma 3.1 we see that the issue is that $\mu$ is unbounded from below. We note that $\mu$ small is exactly the situation in which we would expect a fact rate to be possible (small empirical error). What if $\mu$ is small and bounded from below? (i.e., in a non-interpolation setting.) It seems that this removes this weakness of the prior analysis.

* It seems to me that the bound with binomial upper bound (Lemma A.1) facilitates some information sharing between tasks. On one hand, your analysis recovers the balanced MTL setting so it seems your proof is the correct one. On the other hand, it seems that we're losing the information sharing via Lemma A.1 which seems like a bad idea. Is my intuition about the use of Lemma A.1 incorrect?

* Do the authors have a sense of the geometry of the solution space for the numerical computation in Section 3.4?

* Is there anything technically contrasting between task-centric versus sample-centric versions of the proofs?

* Given Corollary 3.4, in comparison to the numerical numerical details in Section 3.4,  it seems that sample-centric is the nicer version of MTL  versus task-centric. Is that the sense of the authors?

* On line 527 shouldn't it be $B(m_i, \mathcal{R}_i(f_i)/m_i)? As it is doesn't seem to match with  Lemmas A.1-2. This same confusion similarly resides in the application of Lemma A.2 in the final inequality of Equation (53). (Both these also hold for the other proof too. )

**Ethical Concerns:**

["NO or VERY MINOR ethics concerns only"]

**Final Justification:**

I had a productive conversation with the authors wherein my major theoretical concerns were addressed. I didn't trace through every technical detail of the discussion between the AC and the authors. Yet, I see the technical challenges in extending fast rates, at least in the Bayes setting, to unbalanced datasets. As unbalanced datasets are ubiquitous, a dedicated study of these rates is appreciated. As pointed out by the AC, the experiments are at best hard to interpret. In all, I keep my rating. While the contribution of the work towards our understanding in MTL is moderate, illuminating the differences between imbalanced and standard datasets, along with their associated fast rates, is acknowledged.

**Limitations:**

* Task-centric rates for the risks are hard to bound due to the numerical complexity.

**Paper Formatting Concerns:**

* l. 87: I don't believe $\mathcal{M}$ is introduced
* l. 95-102 has several typos: "appliy" "a the"
* Eq. (3) I don't believe the risks indexed by $i$ are introduced.

**Quality:**

2

**Strengths And Weaknesses:**

**Strengths**
* A study of unbalanced datasets in the PAC-Bayes setting is interesting particularly since it appears to differ from the PAC setting, in which it appears to me that everything goes through with standard arguments in the unbalanced setting.
* Their study of task-centric and ample centric is natural and valuable.
* The authors have identified a weakness in the previous arguments for deriving MTL bounds in unbalanced datasets.
* The authors work recovers prior standard bounds in two senses. First, when all the tasks have the same number of samples we recover the balanced fast-rate. Second, the naive approach that would be dependent on the smallest sample size is unable to recover the standard rate (via the Pinsker's inequality trick).  Given these recoveries of prior work it seems that this is the more correct analysis of MTL learning in general and therefore clarifies the literature to the nuance between balanced and imbalanced datasets.
* I appreciated the experiments. In particular Figure 1 that shows the benefits of their method as the task becomes easier which shows the benefit of their approach.
* The discovery that kl-bounds have difficult numerical inversion in the task-centric version.

**Weakness**
* In comparison to prior work, the technical innovation appears to be applying Lemma A.1 to each task independently. Then, after applying Lemma A.2 the harmonic mean is intimidate.
* The difficulty of numerical inversion the task-version appears to limit the applicability of that particular rate (although this may be natural).
* I appreciate the detailed treatment of meta-learning, yet it appears that once the MTL rates are derived the remaining meta-learning analysis is standard.

---

> ### Author Rebuttal · Authors · 2025-07-30
>
> Thank you for your expert review. In the following we hope to answer your questions. We hope it addresses your concern and that you will support our work in the following process.
>
>
>
>
> >  the technical innovation
>
>
> As common in PAC-Bayesian theory, it is not that one specific step of the proof is of high technical depth, but our contribution lies in the choice which individual objects to define and how to combine them in order to achieve optimal rates.
> For Theorem 3.2, we take insight from our lower bound in Lemma 3.1 (which is another technical contribution) and define separate terms for different tasks; however, we then use a shared change of measure step to keep as much information as possible shared among tasks and establish the optimal dependency on the sample sizes. We also bound the moment generating functions of each task separately, because the proof of Lemma 3.1 suggests that a joint bound would lead to a suboptimal rate.
>
>
> > remark about the meta-learning proofs
>
> Indeed, on the technical level, the main challenge of meta-learning bounds is usually to establish the multi-task part. However, we believe the meta-learning bounds are also of interest to the community, and we are not aware of prior work on sample-centric meta-learning at all so far. So, we do not consider including them a weakness.
>
>
> > what if μ is bounded from below?
>
>
> Thank you for the remark. A lower bound on $\mu$ would indeed solve the unboundedness of the MGF. However, as seen in the proof of Lemma 3.1, many terms with inverse dependence on $\mu$ would remain. Therefore, this approach would yield a trade-off between a high lower bound on $\mu$ (which is the object for which we hope to have a small upper bound) or a high upper bound on the MGF. The nice property of the MGF bound we apply is that the dependency of $\mu$ vanishes and we get a term independent of it solely based on $n$ and $m_i$.
>
>
> > intuition about the use of Lemma A.1
>
>
> We would argue that the main source of information sharing is the joint change of measure for all tasks at the same time, which we preserve in our proof (although with the sum of independent terms). The information loss due to the independent use of Lemma A1 and A2 per task can be seen in the additional log terms. It would be interesting to explore if these can be avoided. However, the dominant indicator of information sharing (the KL term) has the correct dependency on $n$ and $m_i$.
>
>
> > geometry of the solution space
>
>
> Given their functional forms, we vaguely think of the Catoni bound as a “linear-ish” upper bound (in particular single-sided), whereas the kl-bound is a “ball-ish” and double-sided. A deeper geometric understanding of the bounds and their interaction could indeed be fruitful.
>
>
> > Is there anything technically contrasting between task-centric versus sample-centric versions of the proofs?
>
>
> The proof of the sample-centric bound is more “standard”: we can bound all tasks with a single term, and we do not face the mentioned limitation on MGF, so it is more straightforward to get the results, and it has a better log dependency. In contrast, in the task-centric setting one has to be more careful in proof steps to avoid the setting of Lemma 3.1, and the fact that split expressions earlier comes at the cost of the additional log terms.
>
>
> > it seems that sample-centric is the nicer version of MTL versus task-centric
>
>
> We believe that the question of sample-centric versus task-centric depends on the application, i.e. which risk reflects the actual object of interest more appropriately. That said, the sample-centric view behaves more similarly to the balanced case, and the analysis is indeed mathematically nicer.
>
>
> > in line 527 shouldn't it be $B(m_i, \mathcal{R}_i(f_i)/m_i)$?
>
>
> We use the notation $B(n,p)$ for a binomial distribution where n is the number of “trials” and p is the “success” probability of each trial. Here we insert the number of samples $m_i$ and their mean $\mathcal{R}_i(f_i)$. The confusion might come from the statement of Lemma A.1. in which there is a term $\mu/t$. Note, however, that in that statement, $\mu$ is the expectation of the sum of $t$ terms, which makes  $\mu/t$ the mean. We will clarify this in the manuscript.
>
>
> > Task-centric rates for the risks are hard to bound due to the numerical complexity.
>
>
> We agree that having this additional optimization step is a bit of a nuisance. However, the actual optimization is possible using standard toolboxes and not actually numerically expensive compared to the model training itself.
>
>
> > Paper Formatting Concerns:
>
>
> Thanks for spotting these, we will fix them.

---

> > ### Comment · Reviewer_4gzb · 2025-08-06
> >
> > Thank you for the detailed response. I found the response clarified my understanding of μ, source of information sharing, and the differences in task versus sample-centric informative. To clarify my initial review, I appreciate the meta-learning content, I was just speaking to corollary nature in respond to additional technical contribution.
> >
> > I also appreciate, and missed during my initial reading, that task-centric risks enjoy relatively inexpensive optimization using standard tools. Along these lines, I agree with Co3j of computational aspects in the appendix receiving more treatment in the body.
> >
> > Speaking to a point I made above,
> >
> > > A study of unbalanced datasets in the PAC-Bayes setting is interesting particularly since it appears to differ from the PAC setting, in which it appears to me that everything goes through with standard arguments in the unbalanced setting.
> >
> > Can the authors remark on if they believe that in the PAC setting there isn't a technical challenge?
> >
> > In fact, prior work gives fast rates in the PAC setting [1, 2] for MTL. To me it appears there is no technical limitation in the unbalanced setting here. Indeed, in [1] it appears that the indexing of sample size plays no major role in the their proofs. Additionally, given the authors detail MTL in the prior work commenting on the differences between the unbalanced setting for PAC and PAC-Bayes can further frame the technical contribution.
> >
> > [1] Yousefi, Niloofar, Yunwen Lei, M. Kloft, Mansooreh Mollaghasemi and Georgios C. Anagnostopoulos. “Local Rademacher Complexity-based Learning Guarantees for Multi-Task Learning.” J. Mach. Learn. Res. 19 (2016): 38:1-38:47.
> >
> > [2] Watkins, Austin, Enayat Ullah, Thanh Nguyen-Tang and Raman Arora. “Optimistic Rates for Multi-Task Representation Learning.” Neural Information Processing Systems (2023).

---

> > > ### Author Response · Authors · 2025-08-06
> > >
> > > Thanks for your response, and sorry we missed responding to this point in our rebuttal.
> > >
> > > Actually, we believe that the difference between unbalanced setting and balanced setting is an intrinsic property of task-centric MTL. We would not assume that it matters much if one uses PAC-Bayes or PAC techniques to characterize generalization. The papers you mention (thank you for bringing these to our attention) indicate at least that the problem is not solved in the PAC literature, either:
> > >
> > > - Paper [1] has only one result for the unbalanced setting: Theorem 1. However, the sample complexity in this theorem depends only on $n_{min}$ (written as $n=\min_t N_t$ there). The emergence of $n_{min}$ seems to stem from the fact that in the proof, upper bounds with identical dependence on the sample size are required for terms of different tasks. However, we are currently not able to say if this is strictly necessary or if the proof could be modified to overcome this. However, with $n_{min}$, the bound’s behavior on the total sample size is clearly suboptimal (the complexity does not improve when tasks get more data, unless $n_{min}$ itself increases), so we would be surprised if the authors had not at least tried to find a better dependence before settling for the published version.
> > >
> > > - Note that the dependency on $n_{min}$ matches exactly our observation after Lemma 3.1, namely that a standard analysis did not suffice to fully characterize the behaviour and complexity of unbalanced MTL.
> > >
> > > - Interestingly, for the rest of the results of the paper, the authors of [1] state: “To present the results in a clear way we always assume in the following that the available data for each task is the same, namely $n$.” The main message of our paper is exactly that the difference of the two settings is not only about simplicity of presentation, and unbalanced MTL needs a more refined analysis to account for different sample sizes.
> > >
> > > - Paper [2] seems to only study the balanced setting, assuming that all tasks have the same sample size $n$ (top of page 5).
> > >
> > > We focused on PAC-Bayes in our paper because of their generality and potential to achieve non-vacuous results for neural networks. However the exact study of the PAC setting could be an interesting direction for future works.
> > >
> > > Thank you for the insightful questions. We will include this discussion, as well as the computational discussion in the manuscript.

---

> > > > ### Comment · Reviewer_4gzb · 2025-08-08
> > > >
> > > > Ah interesting, iI misrecalled the results from [1], I stand corrected, thank you for pointing out the discrepancy. Skimming, it seems that (A.5) in [1] may be the point in which things break down in their proof. I have no remaining questions.

---

### Note · Authors · 2025-08-12

We thank the reviewers and the AC for their time and helpful discussion.

From the discussion phase, we have the impression that we have addressed all the concerns of the reviewers. While surprised about the intense last-minute discussion with the Area Chair, we take from it that the question of fast-rate bounds for unbalanced multi-task learning is relevant and challenging even in the PAC setting, where other tools are available and other challenges emerge${}^1$. We hope that the disagreement with the Area Chair on how to best handle the PAC setting will not negatively affect our chances to present our PAC-Bayesian work to the NeurIPS community.


${}^1$ Out of curiosity, in the meantime we contacted an author of the [Yousefi et al. 2016] paper, who reported that they indeed had unsuccessfully tried to achieve rate that avoid the $m_{min}$ term, and that -- as the suggested by Reviewer 4gzb -- the road block indeed was expression (A.5), i.e., the joint term that includes shared task complexities.

---

### Decision · Program_Chairs · 2025-09-17

**Decision:**

Accept (poster)

**Comment:**

The paper studies fast-rate generalization bounds for multi-task learning (MTL) and meta-learning in the unbalanced setting where tasks have unequal sample sizes. The authors argue that unbalancedness changes the statistical picture relative to the balanced case and that prior proof strategies do not carry over directly. They formalize two distinct settings, task-centric (equal weight per task) and sample-centric (weight by task size), and derive PAC-Bayes KL-style and Catoni-style fast-rate bounds for MTL (and KL-style bounds for meta-learning). Experiments on linear models (MDPR) and low-rank neural networks (split-CIFAR10/100 with ViT features) demonstrate cases where the proposed fast-rate bounds are numerically tighter than standard-rate baselines. The contribution is primarily theoretical; the experiments are illustrative of the bounds’ numerical behavior.

Remaining concerns

1. Related work on fast PAC (non-PAC-Bayes) rates for MTL. The paper completely omits prior fast-rate PAC results for multi-task learning. Because a central question here is what does and does not carry over to the unbalanced setting, a concise survey and a direct comparison would help. In particular, it would be useful to spell out which fast-rate PAC arguments fail (or continue to hold) under task imbalance and why (as also surfaced in the OpenReview discussion).

2. What do the bounds tell us; when is the KL small? The main bounds hinge on the hyper-level divergence KL(Q‖P) between a hyper-posterior and hyper-prior. While the resulting bounds are data-dependent, the paper provides little guidance on when this KL is small in realistic pipelines (e.g., sublinear in the number of tasks n), or how prior/posterior design choices (representations, sharing mechanisms) influence it. At the two extremes, if tasks are independent, the effective complexity decomposes per task, and pooling brings no benefit; if tasks are identical, the favorable KL is unsurprising. What would be an “interesting middle”, a structural characterization of relatedness under which MTL provably helps? In PAC (non-PAC-Bayes) treatments, one often sees explicit notions of task diversity/relatedness leading to rates on the order of 1/(m n) in the balanced case (m samples per task, n tasks). I would even argue that this is what makes the prior work on PAC rates for MTL so helpful and impactful. An analogous insight is not yet articulated here.

3. Purpose and informativeness of the experiments. The bounds require numerical inversion/optimization and do not presently translate into obvious algorithmic choices (e.g., task reweighting, prior design). In the experiments, the hyper-prior is taken to be standard normal, and the resulting comparisons largely mirror how different effective sample sizes behave (e.g., 1/(n·min m_i), 1/(n·m_h), 1/√(n·m_h), where m_h is the harmonic mean of {m_i}). This renders the section largely pointless, merely serving as a check of “bound geometry” rather than an investigation into when the theory is informative for realistic training decisions. The added notion of “simplicity” also appears orthogonal to task-relatedness, so it doesn’t bridge that gap.